# Multimodal determinants of phase-locked dynamics across deep-superficial hippocampal sublayers during theta oscillations

Andrea Navas-Olive[1], Manuel Valero[1], Teresa Jurado-Parras[1], Adan de Salas-Quiroga [1,2], Robert G. Averkin [3], Giuditta Gambino[1,4], Elena Cid[1] & Liset M. de la Prida [1✉]

Theta oscillations play a major role in temporally defining the hippocampal rate code by translating behavioral sequences into neuronal representations. However, mechanisms constraining phase timing and cell-type-specific phase preference are unknown. Here, we employ computational models tuned with evolutionary algorithms to evaluate phase preference of individual CA1 pyramidal cells recorded in mice and rats not engaged in any particular memory task. We applied unbiased and hypothesis-free approaches to identify effects of intrinsic and synaptic factors, as well as cell morphology, in determining phase preference. We found that perisomatic inhibition delivered by complementary populations of basket cells interacts with input pathways to shape phase-locked specificity of deep and superficial pyramidal cells. Somatodendritic integration of fluctuating glutamatergic inputs defined cycle-by-cycle by unsupervised methods demonstrated that firing selection is tuneable across sublayers. Our data identify different mechanisms of phase-locking selectivity that are instrumental for flexible dynamical representations of theta sequences.

---

[1] Instituto Cajal, CSIC, 28002 Madrid, Spain. [2] Department of Biochemistry and Molecular Biology, Instituto Ramón y Cajal de Investigación Sanitaria (IRYCIS) and Instituto Universitario de Investigación Neuroquímica (IUIN), Complutense University, 28040 Madrid, Spain. [3] MTA-SZTE Research Group for Cortical Microcircuits, University of Szeged, Szeged, Hungary. [4] Department of Biomedicine, Neuroscience and Advanced Diagnostics, University of Palermo, Palermo, Italy. ✉email: lmprida@cajal.csic.es

Interaction between internally generated neuronal activity and sensory inputs shapes information processing in the brain[1,2], but similar processes occur at higher-order cortices[3]. At the core of the declarative memory system, the hippocampus acts to integrate behavior and self-generated neuronal sequences in abstract representations[4–8]. While rodents move around in space, hippocampal firing is coordinated by the theta rhythm (4–12 Hz). One mechanism determining translation of behavioral sequences into neuronal sequences is theta phase precession[9,10], which occurs during place field crossing. Hence, cells representing specific locations ahead and behind the subject become assembled to earlier and late cells along theta sequences[4]. Independently of experience, hippocampal pyramidal cells and interneurons fire at preferred theta phases[11–13] and remain locked under the influence of the internal state[14,15]. It is believed that behavioral theta sequences evolve from the transformation of these internal activity patterns into experience-dependent codes[16,17]. However, integrative mechanisms constraining theta phase timing and their dynamical changes during cognitive operations are unknown.

The CA1 region is considered high in the hierarchy of the entorhinal-hippocampal declarative memory system[18]. Traditionally viewed as a single layered structure, recent evidence has disclosed an exquisite laminar organization across deep and superficial CA1 pyramidal sublayers[19,20]. Deep pyramidal cells (closer to the stratum oriens) are more influenced by sensory landmarks while the place fields of superficial cells (closer to the stratum radiatum) appear more context-modulated and abstract[21–23]. In tasks with strong memory demands, such as delayed alternation tasks, firing of deep and superficial CA1 cells can be segregated across theta phases[24,25] and deep cells phase shift specifically during rapid eye movement (REM) sleep[26]. Such a heterogeneous activity indicates that the way different inputs fuse into the existing dynamics may be biased by intrinsic, biophysical, and microcircuit factors. The mechanisms of this high-dimensional integration are unknown.

Here we aim to study factors underlying the internal theta phase preference in the dorsal CA1 using a combination of single-cell and multisite electrophysiological techniques, together with evolutionary computational models. We derive a set of predictions and fill some experimental gaps by exploiting cell-type-specific opto/chemogenetic interventions and unsupervised clustering of theta cycles. We propose that selective phase-locked firing is internally organized in different preferred phases (or firing reservoirs) across deep and superficial CA1 pyramidal cells to build flexible hippocampal codes.

## Results

### Bimodal distribution of theta phase preference across CA1.
To evaluate sublayer-specific neuronal firing we habituated mice to alternate freely between running and immobility in head-fixed conditions (Supplementary Fig. 1). We targeted single CA1 pyramidal neurons juxtacellularly and used linear silicon probe arrays to record extracellular theta oscillations (Fig. 1a, b). Although individual cells tended to fire consistently at specific phases (Fig. 1b), we found a characteristic bimodality at the population level (Fig. 1c; $n = 12$ cells from 12 mice; Rayleigh test $p < 0.05$ for all cells included; mean vector length 0.22 ± 0.03). For individual cells, preferred firing phases were skewed towards the peak (180°) or the trough (0° or 360°) of theta measured at the stratum pyramidale (SP). Single pyramidal cells recorded using similar juxtacellular approaches from rats moving in a familiar open field depicted comparable features ($n = 28$ cells from 28 rats; Fig. 1d, e; mean vector length 0.35 ± 0.15, Raleigh test $p < 0.05$; not significantly different as compared with head-fixed mice). At the population level, bimodality was not dependent on the state (awake versus REM sleep) or on the type of theta (running vs. attentional, whisking, or grooming theta; Supplementary Fig. 2A–E), even though some cells shifted phase across conditions, as previously reported[26] (Fig. 1f).

We then looked for other explanatory variables. A multivariate analysis (Harrison–Kanji test, equivalent to a circular two-way analysis of variance (ANOVA)) of the subset of morphologically identified cells demonstrated statistical effects for sublayer ($\chi^2(2) = 10.7$, $p = 0.0136$) and preparation ($\chi^2(2) = 8.5$, $p = 0.0046$; $n = 10$ cells from 10 mice and $n = 20$ cells from 20 rats) (Fig. 1g). In general, superficial cells tended to fire closer to the theta trough and deep cells more towards the theta peak, with sharp differences in head-fixed conditions (Fig. 1g). The effect of the preparation may result from inter-species differences, but also from the influence of different input pathways in head-fixed vs. freely moving behavior (i.e., vestibular and head-direction signaling, optic flow, etc.)[27]. No differences in the local theta could explain results (Supplementary Fig. 2F). No additional behavioral factors were observed to explain population bimodality (Supplementary Fig. 1G), suggesting it emerges from the internal microcircuit organization across deep and superficial sublayers.

To further confirm cell-type specificity of phase-locking distribution we performed optogenetic tagging of deep and superficial CA1 pyramidal cells (Fig. 1h). We used Calb1-cre mice injected with an adeno-associated virus (AAV5) to restrict expression of channelrhodopsin2 (ChR2) and the yellow fluorescent protein (YFP) in a subset of superficial CA1 cells (30% of Calbindin+ cells were YFP+), whereas deep cells were targeted in the transgenic line Thy1-ChR2 (ref. [28]) (75% of Calbindin− cells). Using integrated micro-light-emitting diode (LED) optoelectrodes, we exploited nanowatt blue light stimulation for controlled opto-tagging of deep and superficial cells[29]. Limited connectivity between CA1 pyramidal cells exclude potential indirect excitatory effects[30,31], but see ref. [32]. We successfully isolated $n = 7$ superficial and $n = 14$ deep pyramidal-like units that were significantly theta modulated, while head-fixed mice run (Rayleigh test $p < 0.05$) (Fig. 1i). Consistent with juxtacellular data, we found significant segregation of deep and superficial firing across theta phases (F(1,20) = 17.7, $p = 0.0005$; Watson–Williams test, equivalent to a circular one-way ANOVA), further reinforcing the idea that population bimodality is determined by different cell-type contribution.

Thus, single-cell experimental data obtained with different approaches and in different species demonstrate that under basal conditions dominated by the internal dynamics, the firing of deep and superficial CA1 pyramidal cells is segregated during theta oscillations.

### Computational models tuned with evolutionary algorithms.
In order to understand the underlying mechanisms, we implemented a biophysically realistic model of CA1 pyramidal cells that included known excitatory and inhibitory inputs, using morphological reconstructions from a public database (http://neuromorpho.org/) and the Hodgkin–Huxley multi-compartment formalism in the Neuron + Python (Fig. 2a; see Methods). Our simulations were based on a realistic full-scale model of the CA1 microcircuit able to autonomously generate theta oscillations[33].

To constrain the parametric space (23 free parameters, listed in Supplementary Table 1), we adopted genetic algorithms (GAs) to identify values for passive, active, and synaptic conductances resulting in realistic behavior in a given morphology[34,35]. To this purpose, the effective conductance of each channel fitted by GA was set as the product of the maximal conductance (Gmax) and a GA factor. We tuned GA factor values in a range so that the

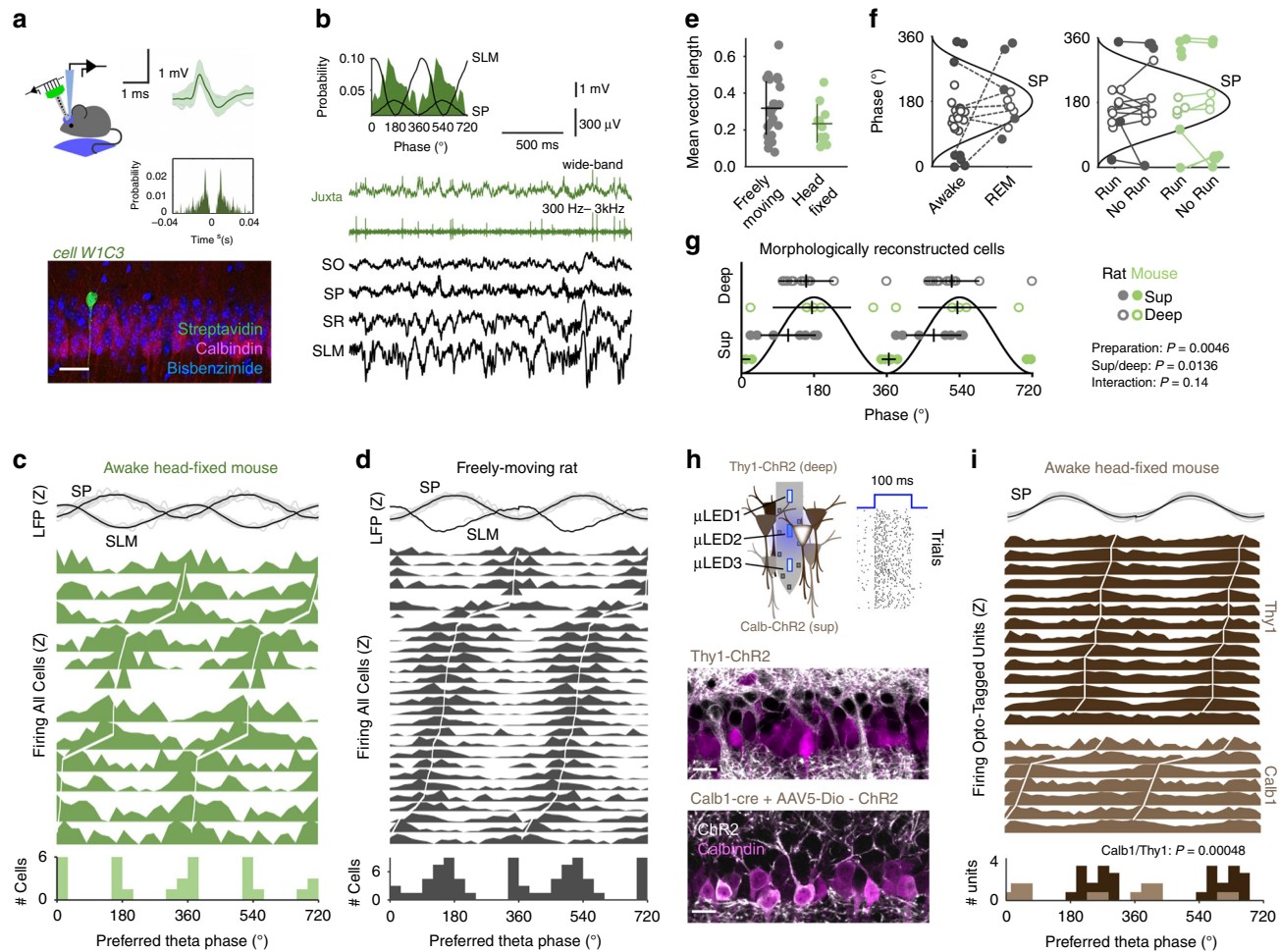

**Fig. 1 Bimodal distribution of theta phase-locked firing across CA1 sublayers. a** Representative example of a CA1 pyramidal cell recorded juxtacellularly from a head-fixed running mouse. See average action potential waveform and autocorrelation at right, and morphological identification as deep cell at bottom. Scale bar 50 μm. **b** Theta phase-locked firing histogram of the cell shown in **a**. Raw local field potential (LFP) traces and juxtacellular signals are shown below: SO, stratum oriens; SP, stratum pyramidale; SR, stratum radiatum; SLM, stratum lacunosum molecular. **c** Theta phase firing histogram from single cells recorded in awake head-fixed mice, ranked according to their preferred phase ($n = 12$ cells). Individual (gray) and mean (black) theta filtered LFP signals at top are z-scored (Z). The population histogram at bottom represents the distribution of mean preferred phases from individual cells. **d** Same for single pyramidal cells recorded juxtacellularly in freely moving rats ($n = 28$ cells). **e** Individual and mean ± SD values of vector length per cells reported before in head-fixed and freely moving conditions. No differences between groups. **f** Awake/REM sleep (left) and RUN/no-RUN (running vs. other motor behavior) dependency of theta phase preference of single pyramidal cells. See legend in (**g**). **g** Multivariate analysis of morphologically identified cells demonstrate effects per location (Harrison–Kanji test for deep-superficial location: $\chi^2(2) = 10.7$, $p = 0.0136$) and preparation ($\chi^2(2) = 8.5$, $p = 0.0046$); no interaction ($\chi^2(1) = 2.1$, $p = 0.148$). Phase preferred data from each morphologically identified cell is depicted separately for deep and superficial cells in both preparations. **h** Optogenetic tagging of single units using micro-LED optoelectrodes in two transgenic lines allowed identifying superficial (Calb1-cre + AAV5-DIO-ChR2-YFP) and deep CA1 pyramidal cells (Thy1-ChR2). Raster at right shows response of one representative unit to 79 trials of nanowatt blue light stimulation in a Thy1-ChR2 mouse. Bottom, ChR2 signal (white) and Calbindin immunostaining (magenta) are shown in false colors (one confocal plane from each line). Scale bar 25 μm. **i** Phase-locked firing histograms from Calbindin+ superficial ($n = 7$ units from 3 mice) and Thy1+ deep opto-tagged units ($n = 14$ units from 2 mice; statistically different Watson–Williams test, F(1,20) = 17.7, $p = 0.00048$. Note that bimodal distribution of preferred theta phases can be explained by different cell-type contribution.

model cell targets intrinsic (Fig. 2b) and synaptic experimental responses (Fig. 2c, Supplementary Fig. 3A, and Table 2). Passive parameters were fitted similarly (Supplementary Tables 1 and 2). This strategy provides a set of parameter values that successfully fit the model behavior. For instance, more than 4000 intrinsic sets of GA factors fitted experimental somatic and dendritic responses to current pulses (Fig. 2b; black dots), providing individuals (i.e., models) with some level of intrinsic heterogeneity. Similarly, experimental synaptic responses to CA3 input stimulation were used to fit the proportion of CA3 synapses as well as those GABAergic synapses activated feedforwardly, providing additional heterogeneity (Fig. 2c; >5000 sets of synaptic GA factors).

Values for the remaining parameters not fitted by GA were chosen randomly from known experimental ranges (Supplementary Table 1, Supplementary Table 2 for intrinsic parameters, and Supplementary Table 3 for synaptic parameters).

To evaluate the space of possible GA models, we adopted the following validation strategy. First, we fitted intrinsic parameters to a given morphology as previously explained (e.g. n128 from the Neuromorpho Turner archive) and propagated them to different morphologies (n127 and n409 from the Turner archive and sup1-I040913C2 from the Prida archive) by randomly selecting 20 intrinsic individuals (Supplementary Fig. 3B). We then validated those individuals that successfully fitted the experimental target

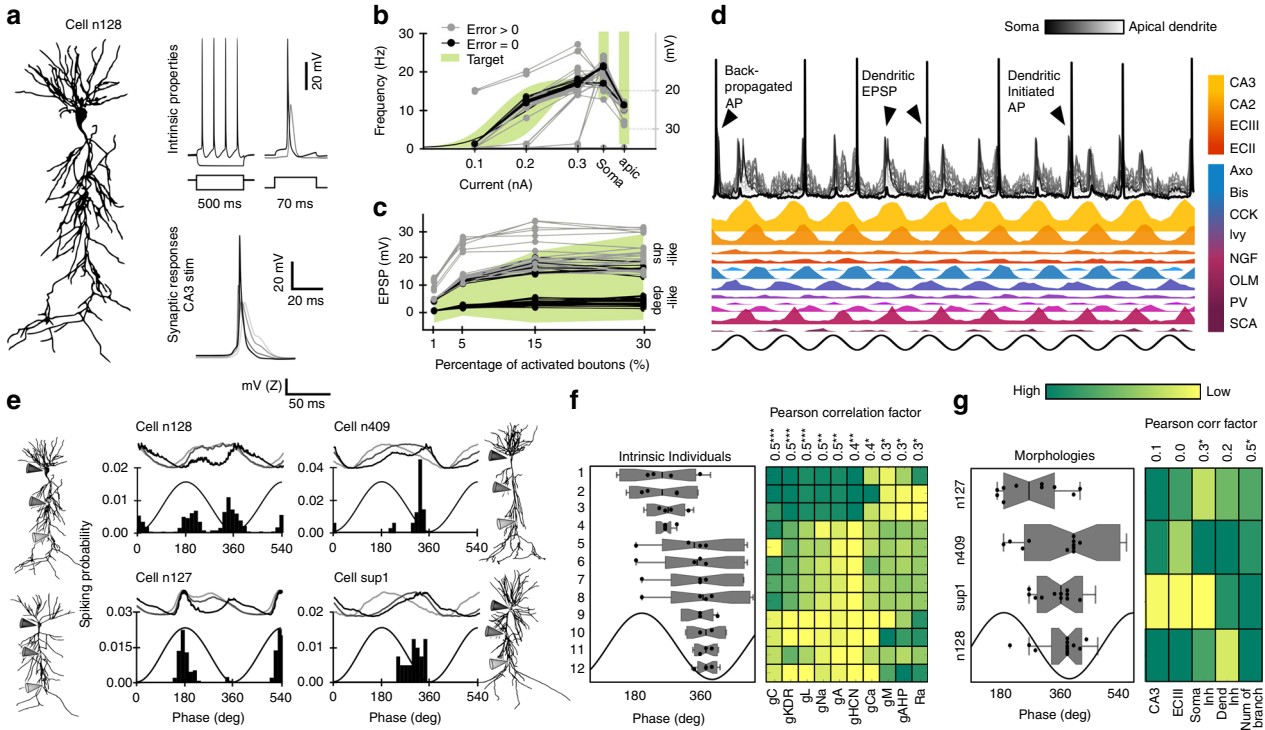

**Fig. 2 Genetically constrained models of CA1 pyramidal cell activity. a** Activity of morphologically identified CA1 pyramidal cells was modeled using the multi-compartment Hodgkin–Huxley formalism. **b** Genetic algorithms restricted the intrinsic parameter space (GA factors) by fitting experimental somatic and dendritic responses to current pulses (green). Different combination of GA factors (individuals) giving experimentally valid input-output responses (Error = 0 in target; n = 45 individuals) are shown in Supplementary Fig. 3C. **c** Same as in (**b**) for fitting synaptic responses to Schaffer collaterals stimulation (n = 60 individuals). See valid GA factors and examples in Supplementary Fig. 3D. Note synaptic responses similar to those recorded in deep and superficial CA1 cells in vivo[36]. **d** Theta-modulated activity simulated in the synthetic cell shown in (**a**) in response to a biologically realistic collection of glutamatergic (CA3, CA2, ECIII, and ECII) and GABAergic (Axo, Bis, CCK, Ivy, NGF, OLM, PV, and SCA) theta-modulated inputs (Supplementary Table 3). **e** Theta phase-locking dynamics of same set of intrinsic features (individual #5) in four different morphologies. Upper traces show membrane potential at the soma (black), proximal (gray), and distal dendrites (light gray). Data from 1000 theta cycles. **f** Effect of individual variability of intrinsic properties. Data were ranked according to results from a permutation test that maximized correlations between individuals and phase preference. Mean data from each intrinsic individual in the four morphologies is represented as box-whisker plots. The number of dots per individual varies according to inclusion criteria (individuals with realistic firing rate values in at least three out of four morphologies, n = 3 for individuals 9–11, n = 4 for the rest). The matrix at right shows mean GA factors across individuals. Note Pearson's correlation coefficient R and significance at top: gC (p = 0.0003), gKDR (p = 0.0002), gL (p = 0.0006), gNa (p = 0.002), gA (p = 0.002), gHCN (p = 0.003), gCa (p = 0.02), gM (p = 0.05), gAHP (p = 0.02), and Ra (p = 0.03). **g** Same as in (**f**) for morphologies. The number of dots per morphology varies according to inclusion criteria: n127 (n = 10 synthetic cells), n409 (n = 11), sup1 (n = 11), and n128 (n = 12). Note significant effect of perisomatic inhibition (p = 0.041) and the total dendritic branches (p = 0.013). ***p < 0.001; **p < 0.01; *p < 0.05.

across all morphologies and replaced non-valid individuals by new random sets from the pool until the full dataset was validated (Supplementary Fig. 3B; upper part). Next, we proceed similarly for synaptic GA factors (Supplementary Fig. 3B; bottom). By allowing synaptic traits to fit independently across different morphologies we gained additional heterogeneity to be exploited for analysis. An alternative strategy to evolve GA for fitting all morphologies together provided less heterogeneity at higher computational cost. Importantly, the resulting synthetic cells (i.e., combination of intrinsic GA factors, synaptic GA factors, and morphology) accounted well for other non-fitted intrinsic properties of CA1 pyramidal cells, such as spike hyper- and after-depolarization, sags, etc. (Supplementary Fig. 3C right and Supplementary Table 4). Similarly, we noted that for some morphologies the GA converged spontaneously on low and high-amplitude synaptic potentials, providing sets of synthetic cells responding similar to CA3 stimulation than deep and superficial CA1 cells in vivo[36] (Fig. 2c and Supplementary Fig. 3D).

We next evaluated synthetic cells independently on GA fitting by simulating theta phase-locking firing. To this purpose, we submitted them to a biologically realistic collection of

glutamatergic and GABAergic theta-modulated inputs (Fig. 2d and Supplementary Table 3 and references therein; 1000 cycles per simulation and cell). Accordingly, synthetic cell behavior was theta modulated, with notable cycle-by-cycle variations of dendritic and somatic activity caused by small random fluctuations of GABAergic and glutamatergic inputs (Fig. 2d arrowheads and see oscillatory properties in Supplementary Table 4). Interestingly, we noted effects of both morphologies and intrinsic variability on firing phase distribution. For instance, a given intrinsic individual expressed in different morphologies showed different phase-locking preference and membrane potential dynamics (Fig. 2e). They also exhibited variability in terms of firing rate (Supplementary Fig. 3E), similar to real CA1 neurons[26] (Fig. 1).

To understand this point better, we evaluated the effect of GA factors by choosing only synthetic cells with realistic firing rate values (0.01–8 Hz) in at least three out of four morphologies (Supplementary Fig. 3E). We noted different trends of synthetic cells to phase-lock across individuals and morphologies. For instance, individual #10 fired at very similar theta phases when expressed in different morphologies, whereas individual #1

exhibited large dispersion (Fig. 2f). This is because some intrinsic factors, such as the axial resistance and the maximal conductance of sodium and A-type potassium channels, make synthetic cells firing more or less reliably to dominant inputs. Consequently, expression level of some passive and active intrinsic properties had significant correlation with phase preference (Fig. 2f, matrix at right).

We next evaluate the effect of morphological features on phase preference by pooling data from all individuals in a given morphology (Fig. 2g). In general, morphologies n409 and n127 showed a trend for some cells with deep-like traits (firing towards the peak) versus n128 and sup1 (a superficial cell reconstructed in our lab) with more consistent superficial-like firing (around the trough) (Fig. 2g). Interestingly, at the population level, we noted a bimodal distribution of the preferred firing phase from synthetic cells in some morphology (Supplementary Fig. 3E, bottom plot; e.g., n409 and n127). Thus, we analyzed potential interactions between morphology and intrinsic factors using a two-way ANOVA and found statistical effects for the following set of conductances: type A ($F_{(23,9)} = 11.3$, $p = 0.0003$), type C ($F_{(28,2)} = 26.4$, $p = 0.037$), type L calcium channel ($F_{(12,24)} = 26.4$, $p = 0.037$), HCN channel ($F_{(9,28)} = 2.8$, $p = 0.017$), type M ($F_{(11,25)} = 2.2$, $p = 0.044$) and the leak channel ($F_{(9,28)} = 2.8$, $p = 0.017$). Although phase preference across morphologies did not correlate with the distribution of glutamatergic inputs from CA3 and ECIII, we found significance for perisomatic inhibition and the total dendritic branches (Fig. 2g right). Thus, we hypothesized that variations in the distribution of synaptic inputs and intrinsic properties in interaction with morphological features play key roles in determining firing dynamics across CA1 sublayers.

**Major role of cell-type-dependent perisomatic inhibition**. Deep and superficial CA1 pyramidal cells are unequally innervated by some GABAergic interneurons[36,37], with about 70% difference in the density of perisomatic contacts from PV+ (higher in deep) and CCK+ (higher in superficial) basket cells[36]. We built deep-like (blue) and superficial-like (magenta) synthetic cells by readjusting the proportion of PV and CCK perisomatic boutons accordingly (Fig. 3a; expressed as the percentage SP boutons detailed in Supplementary Table 3). Such a simple manipulation had strong consequences in theta phase-locking dynamics. Tuning spikes to the theta peak or trough depended on a net phase-specific depolarization measured at the soma, which could advance or delay firing (Fig. 3a, membrane potential traces at right-hand). We thus examined the effect of completely removing GABAergic synapses and found multiple windows of opportunity for phase-locking firing shaped by glutamatergic inputs from the peak (ECIII) to the trough (CA3, CA2, and ECII) (Fig. 3a bottom). As PV+ and CCK+ basket cells fire at opposite theta phases[11,12], a 70% reduction of synapses from one or another was enough to bias firing in response to different glutamatergic input pathways.

To evaluate robustness of this result across individuals and morphologies, we unfolded the original population of synthetic cells in deep-like and superficial-like cells (Supplementary Fig. 4A, B; $n = 48$ synthetic cells; 12 individuals per morphology). A trough-peak (TP) index captured different trends (Fig. 3b; sublayer-like effects in a one-way ANOVA, $F_{(2,108)} = 41.9$, $p < 0.0001$ for the total dataset). We found similar results running the full model in supercomputer clusters of the Neuroscience Gateway (https://www.nsgportal.org/), where theta oscillations emerged autonomously (Supplementary Fig. 5)[33]. Strikingly, the effect was dependent on the original phase preference determined by the GA fitting and morphology. For instance, most original

individuals with morphology n128 behaved as superficial CA1 pyramidal cells (TP index > 0) and therefore the effect of decreasing 70% PV+ boutons (making them more superficial; magenta) was nonsignificant in contrast to the effect of decreasing CCK+ boutons, which made them deep-like (blue, TP index < 0; Fig. 3b, all significant paired $t$-tests, $p < 0.0001$). This suggests that the ability of a particular interneuron to shift firing of pyramidal cells will depend on their basal theta phase preference. We confirmed this idea by testing the effect of 70% reduction from all other GABAergic types and predict significant effects of bistratified and oriens-lacunosum moleculare (OLM) interneurons only for deep-like cells (Supplementary Fig. 4C, D). Importantly, reducing connectivity (synaptic conductance) or the corresponding input rate (same conductance but lower pre-synaptic firing rate) gave comparable results in all our simulations, suggesting that the mechanism could be either hardwired or dynamically regulated.

To make more testable predictions, we simulated the effect of fully removing PV + basket cell inputs and confirmed that most synthetic neurons preferentially shifted to the falling theta phase (Supplementary Fig. 6A, $n = 32$ synthetic cells taken randomly from the deep- and superficial-like pool; 8 individuals per morphology). To test this result experimentally, we injected PV-Cre mice with AAV to express the inhibitory designer DREADD receptor hM4D(Gi) coupled to mCherry ($n = 7$ mice). As PV is expressed in many GABAergic populations, we evaluated co-localization of mCherry with complimentary immunohistochemistry and found specific expression in subsets of interneurons (Fig. 3c). Hence, we updated our computational predictions accordingly (Fig. 3d) and checked for effects of clozapine-$N$-oxide (CNO), the specific ligand of hM4D(Gi) receptors. Given that intraperitoneal injections compromised single-cell recordings, we evaluated theta phase preference mostly after CNO injections. In a subset of experiments with enough stability to guarantee longitudinal analysis, we confirmed lower theta power (paired $t$-test, $t_{(4)} = 3.9975$, $p = 0.016$; $n = 5$), similar extracellular phase reversal profiles (Supplementary Fig. 6B; no effect of vehicle tested separately in $n = 2$ sessions) and significant reduction on the firing rate of putative PV+ basket cells after CNO (Supplementary Fig. 6C, D).

Consistent with simulations we found that single pyramidal cell firing recorded after CNO in PV-hM4D(Gi) mice concentrated at the falling theta phase using both juxtacellular ($n = 5$ pyramidal cells) and multisite sorting methods ($n = 23$ putative pyramidal cells, Fig. 3e). Phase-locked firing distribution after CNO was similar to simulation predictions (circular Watson–Williams multi-sample test, $F_{(1,53)} = 0.02$, $p = 0.88$), but different to control distributions for identified deep ($F_{(1,44)} = 4.4$, $p = 0.04$) and superficial cells ($F_{(1,36)} = 4.6$, $p = 0.038$) reported in Fig. 1. Moreover, the subset of putative pyramidal units followed longitudinally along the experiment shifted consistently after CNO ($n = 12$, Fig. 3f) so that theta phase distribution before and after was significantly different (Supplementary Fig. 6E; $F_{(1,22)} = 4.5$, $p = 0.046$). Interestingly, phase-locked dynamics become indistinguishable across sublayers after CNO in PV-hM4D(Gi) mice but not after vehicle nor in wild-type mice treated with CNO, confirming specific effects (Supplementary Fig. 6F, G; see statistical comparisons in caption). Therefore, dedicated GABAergic micro-circuits constrain phase selection of subsets of CA1 pyramidal cells during theta oscillations.

**Contribution of glutamatergic inputs across CA1 sublayers**. Next, we aimed to examine the role of the glutamatergic input pathways in shaping phase preference across sublayers. Given the major role of CA3 and ECIII inputs, we first focused on

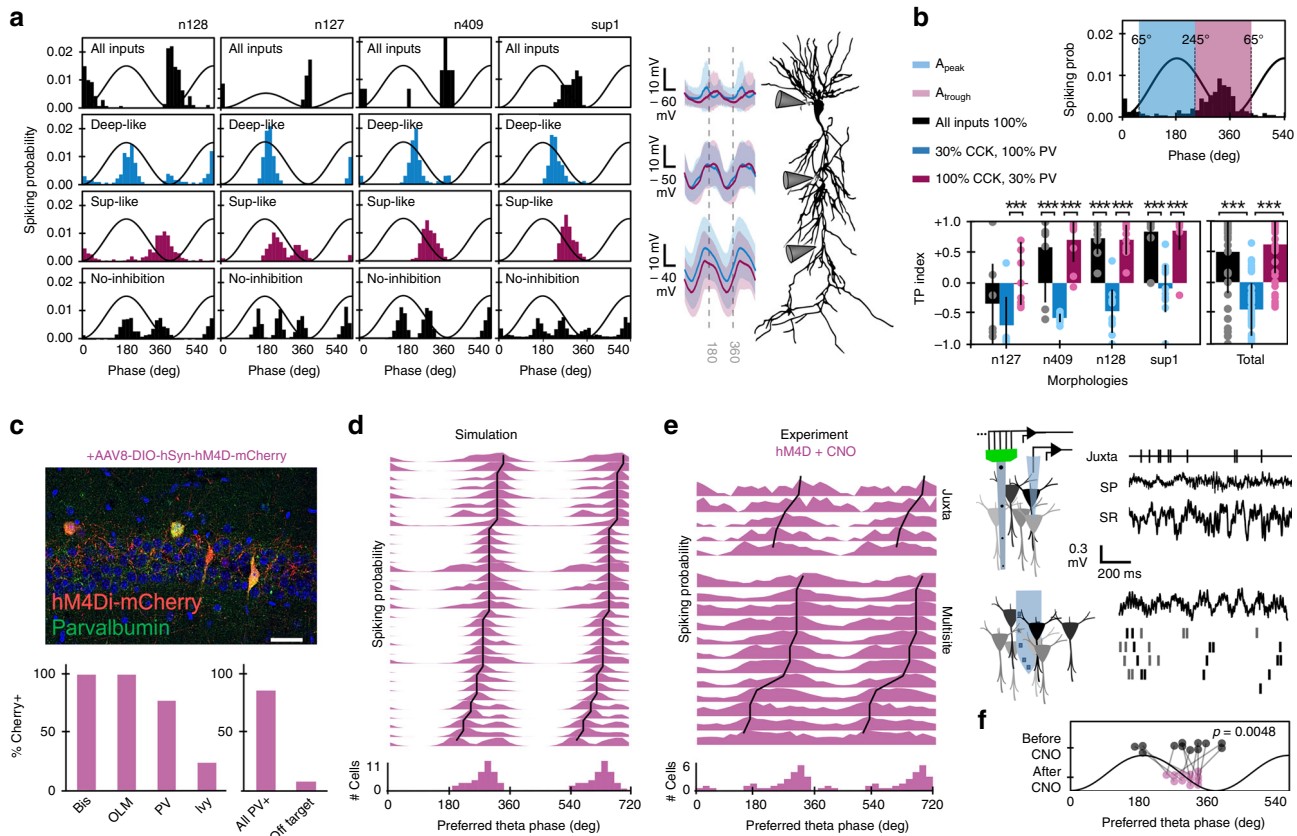

**Fig. 3 Interneuron-specific control of theta phase preference. a** Effect of different distribution of perisomatic GABAergic inputs from PV and CCK basket cells. Note reliable phase-locking behavior determined by inputs typical of deep (30% CCK, 100% PV; blue) and superficial (100% CCK, 30% PV; magenta) pyramidal cells, consistent with experimental data[36]. Data from 1000 theta cycles (individual #12 in four morphologies). Bottom histograms show effect of fully removing inhibition. Traces at the right-hand side show mean membrane potential dynamics at different somatodendritic compartments of deep- (blue) and superficial-like (magenta) synthetic cells shown at left. **b**, Effect of different PV/CCK inputs on 12 individuals from the four different morphologies ($n = 48$ synthetic cells), quantified by the TP index. Individual and mean $+/-$ SD values are shown. Data tested with paired t-test for deep- versus superficial-like groups within each morphology. Full dataset tested with one-way ANOVA, F(2,108) = 41.9 and post-hoc t-tests. ***$p < 0.0001$ (from left to right: $p = 6 \times 10^{-5}$, $p = 10^{-4}$, $p = 8 \times 10^{-7}$, $p = 5 \times 10^{-5}$, $p = 4 \times 10^{-5}$, $p = 3 \times 10^{-6}$, $p = 4 \times 10^{-7}$). **c** Chemogenetic approaches allowed to test predictions on the effect of silencing PV+ interneurons in PV-Cre mice injected with hM4D(Gi). Immunohistological validation of interneuronal cell-types affected by hM4D(Gi): Bistratified cells (Bis, $n = 2$), oriens-lacunosum moleculare cells (OLM, $n = 4$), PV basket cells or axo-axonic cells (PV, $n = 23$), Ivy cells ($n = 12$), All PV+ ($n = 42$), and Off target ($n = 4$). Data from 3 sections from a representative mouse. Scale bar 50 μm. **d** Model prediction of the population level effect of blocking GABAergic inputs from PV basket cells, bistratified cells, OLM interneuron, and a minority of Ivy cells ($n = 32$ synthetic cells; 1000 cycles).
**e** Examples of chemogenetic effects and histogram quantification of the full dataset ($n = 5$ juxtacellular; $n = 23$ single units from multisite recordings). Note similar distribution as compared with simulations (circular Watson–Williams multi-sample test, F(1,53) = 0.02, $p = 0.88$). All cells/units significantly theta modulated as tested by Rayleigh (p < 0.05). **f**, Change of theta phase preference in a subset of $n = 12$ putative pyramidal cells recorded before and after CNO in PV-hDM4(Gi) mice. Significant differences, circular two paired sample test, t(11) = 5.7846, $p = 0.0048$.

understanding how individual cells integrate their rhythmic synaptic potentials arriving at the falling theta phase and peak, respectively. To test for the effect in a comparable number of theta cycles, we chose GA-fitted values corresponding to larger and smaller synaptic conductances on either pathway in $n = 48$ synthetic cells.

As previously suggested[38,39], we found that entorhinal and CA3 inputs could interact nonlinearly depending on their relative strength and the dominant type of phase-dependent inhibition (Fig. 4a). For low ECIII inputs, somatic depolarization was dominated by CA3 inputs and synthetic cells fired preferentially towards the theta trough independently on the PV/CCK axis (Fig. 4a, bottom). In contrast, for intermediate and high ECIII activity levels, distal and proximal dendritic potentials interacted with perisomatic inhibition to shape firing preference (Fig. 4a, middle and top). This was clearly visible in membrane potentials at the main proximal and distal dendritic trunk (Fig. 4b, gray traces). Strikingly, bimodality of phase-locked firing could not be

explained by the phase of the maximal dendritic depolarization alone, even under conditions of maximal inputs on either pathway (Fig. 4c, arrowheads). This further supports the idea that phase-locking firing is the result of several interacting factors.

To identify multimodal determinants of phase selection across individual cells we ran a logistic regression analysis that included all different explicative morphological, biophysical, intrinsic and synaptic input features from a big dataset ($n = 731$ synthetic cells × 97 features, >300,000 theta cycles; Supplementary Fig. 7A). We evaluated the effect of up- and downregulations of each feature in determining phase preference and selected those meeting significance in circular Rayleigh statistics (Supplementary Fig. 7B). Consistent with results above, we found that PV, CCK, and ECIII inputs largely explained phase bimodality across the population of synthetic cells, but other factors emerged as well generating additional predictions (Fig. 4d). For instance, having lower or higher number of basal dendrites had contrasting effects on phase preference, in contrast to the number of apical dendritic

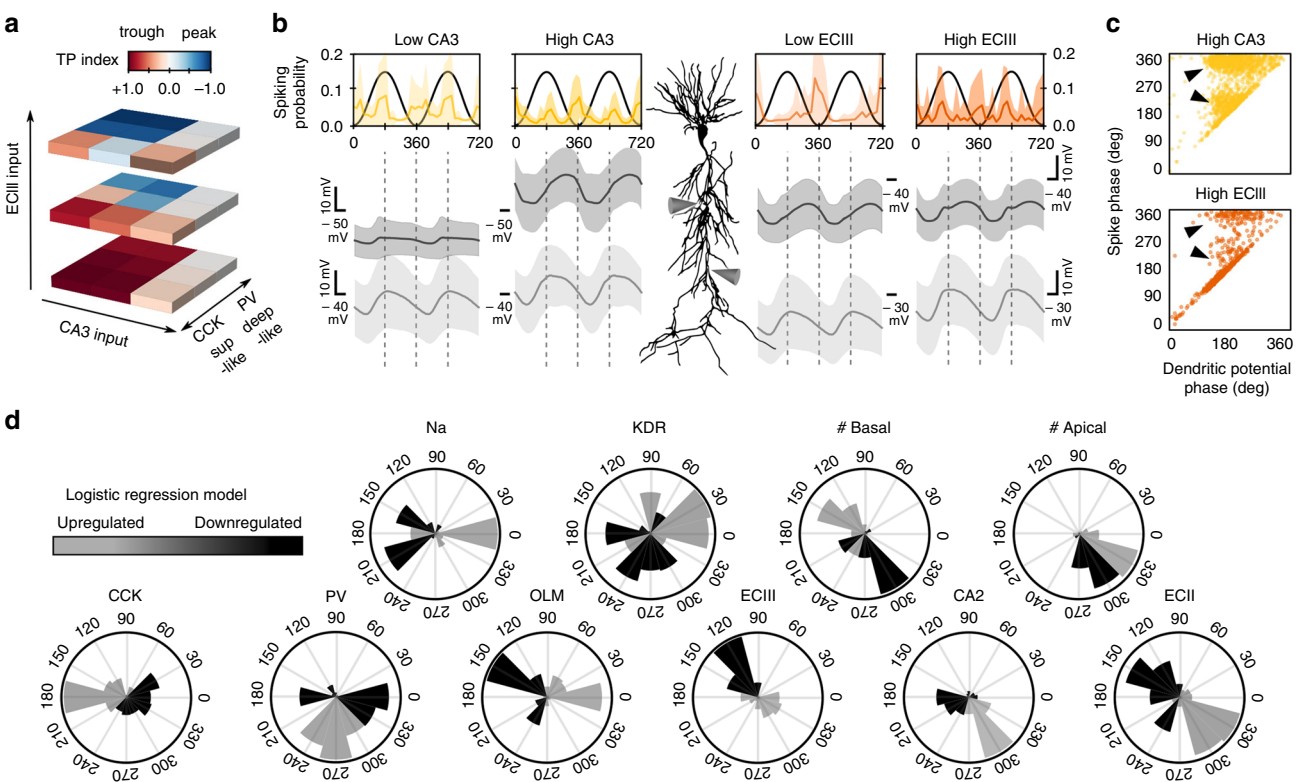

**Fig. 4 Effect of glutamatergic input pathways. a** Effect of CA3 and ECIII input pathways on phase preference of synthetic cells across the PV/CCK axis. **b** Mean phase-locked firing histograms and membrane potential dynamics at the proximal and apical dendritic trunk from all synthetic cells examined under low and high levels of activity of the CA3 (yellow) and ECIII inputs (orange). Data from 48 synthetic cells in each condition (1000 theta cycles each). **c** Relationship between the firing phase and the phase of the maximal dendritic depolarization measured at the proximal apical trunk. Each dot represents data from one spike from 48 synthetic cells in the high CA3 and high ECIII conditions. Arrowheads indicate uncorrelated phase-locking dynamics between firing and dendritic potentials. Bimodality was stronger during higher CA3 inputs. Data from 1000 theta cycles in each condition. **d** Results of a multinomial logistic regression model suggest phase-specific contribution of different individual factors. Black, upregulation. Gray, downregulation. Data from 731 heterogeneous synthetic cells and >300,000 theta cycles. See Supplementary Fig. 7.

branches which could explain only minor phase difference (Fig. 4d). Notably, superficial and deep pyramidal cells differ in the number of basal and apical dendritic branches[40], supporting cell morphology as a critical factor as suggested above. Similarly, we found that inputs from CA2 and ECII contribute to set neuronal firing along the raising and the falling theta phase. We confirmed one of these predictions by running a new set of simulations at different levels of CA2 inputs (Supplementary Fig. 7C). Other predictions include asymmetric effects of the up and down-regulation of entorhinal inputs and the effect of OLM interneurons, which was supported by independent simulations (Supplementary Fig. 4C). Finally, the regression analysis also pointed to some intrinsic factors influencing phase-locking behavior (Fig. 4d and Supplementary Fig. 7B, D). Because of the distribution of some of these factors follows deep-superficial gradients[41], these results provide additional support to the idea that under basal conditions selective phase-locked firing is differentially organized across sublayers.

**Dynamic reorganization of phase-locked firing.** An additional corollary of our simulations is that phase-locked firing could be tuned from preferred phases provided physiological mechanisms operate to regulate the relative contribution of some factors. For instance, simulations in Fig. 4a predict that transient increases of entorhinal inputs could act to shift firing differently from the preferred phases established along the deep-superficial PV/CCK axis. In vivo, activity levels at CA3 and ECIII synapses are regulated by the cognitive load[24,42]. Theta cycles with low beta

(20 Hz) and gamma (30–60 Hz) transient spectral components (tSC1 and 2, respectively) are associated with stronger CA3 inputs, whereas cycles with mid (tSC3) and higher gamma components (tSC4) are rather reflecting instances of stronger entorhinal inputs[42]. To better understand whether these mechanisms can cooperate to advance or delay firing on demand, we exploited the spectral variability of theta-nested oscillations as a proxy for dynamical changes of entorhinal and CA3 inputs.

Using self-organizing maps (SOMs)[43], we first categorized waveforms of individual cycles in different classes (Fig. 5a). Then for the separate tSC1-2 (high CA3) and tSC3-4 (high ECIII) classes we built theta phase histograms of single cells recorded simultaneously by juxtacellular and multisite methods (Fig. 5a, Supplementary Fig. 8A, B, and see Methods). To ensure robust statistics across different theta classes, data from each cell/unit was contrasted against 1000 surrogates and only those meeting significance (Rayleigh test and TP index) were further considered (Fig. 5b and Supplementary Fig. 8C; $n = 7$ juxtacellular from 7 mice, 4 deep, and 3 superficial; $n = 96$ units from multisite recordings, 3 sessions from 3 mice, 70 deep, and 26 superficial). In addition, to facilitate interpreting results at the population level and to further constraint chance effects, we restricted analysis to units with consistent coactivation during cycles, as tested with a rank order test[44]. To this purpose, for all cycles of a class the spike timing of individual units was ranked from 0 to 1 (rank order). Significant rank order distributions were identified using pairwise correlations between all cycles and tested against the shuffle distribution (Fig. 5c; pairwise correlations > 0.8; 500 shuffles). This

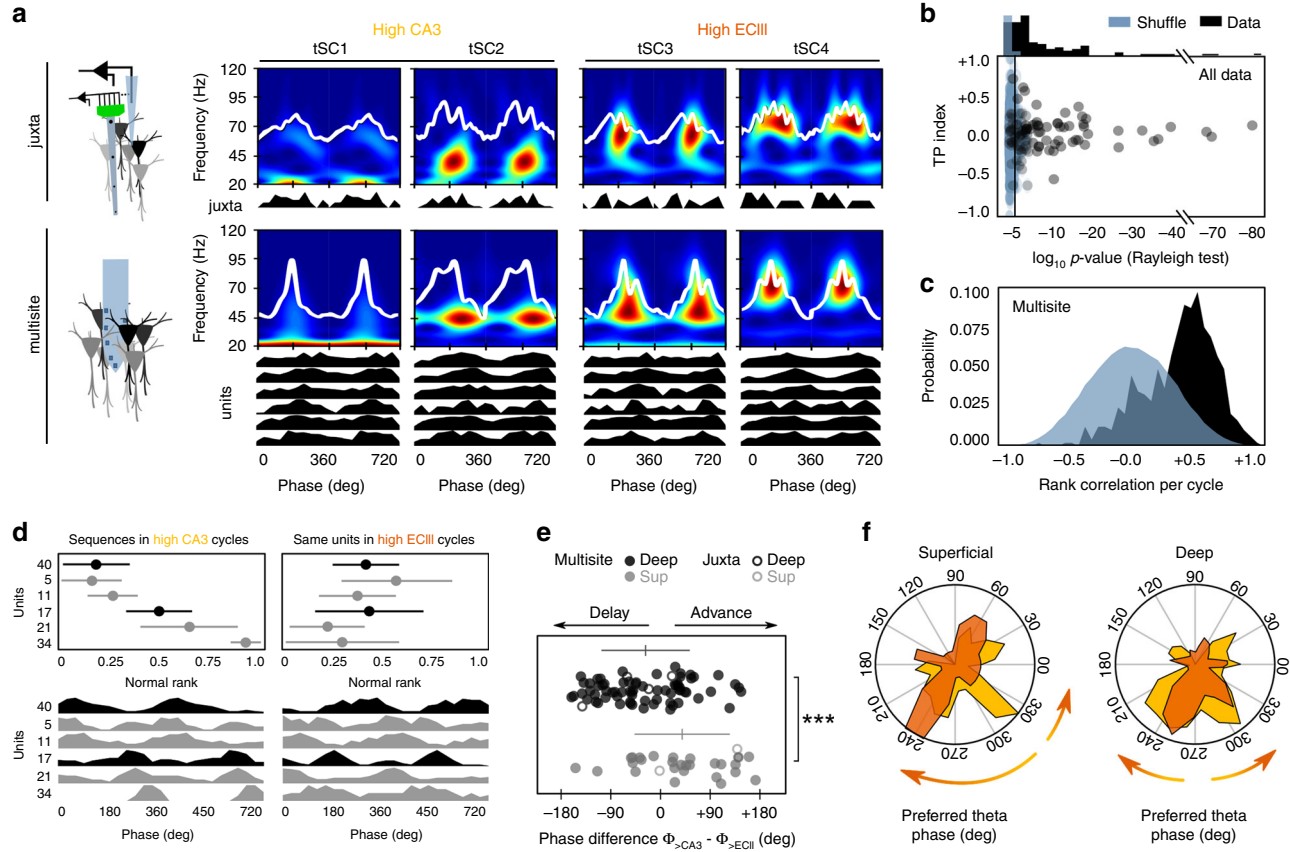

**Fig. 5 Dynamic phase shifts of deep and superficial CA1 pyramidal cells. a** Unsupervised classification of theta cycles in different classes allowed evaluating changes of firing dynamics from single-cells recorded juxtacelullarly and using multisite probes. Cycles classified as tSC1 and 2 are associated to larger CA3 inputs (yellow) while tSC3 and 4 reflect larger entorhinal layer III inputs (orange). **b** Results from the surrogate tests (1000 shuffles) on significant theta modulation of single-cell data during theta cycles subclasses. Only cells/units meeting significance ($p < 0.05$ Rayleigh test) for both tSC1-2 and tSC3-4 firing histograms were considered for further analysis ($n = 7$ juxtacellular cells, $n = 96$ units from multisite recording). **c** Rank order tests were used to identify sequences of units repeating consistently across classified theta cycles as tested against chance order (500 shuffles). **d** Example of the effect of high ECIII versus high CA3 inputs in deep and superficial units firing together over several theta cycles. Upper plots show mean rank order ± SD for each unit for all high CA3 (from up to bottom: 19, 13, 17, 17, 18, and 10 cycles) versus high ECIII cycles (from up to bottom: 13, 15, 13, 12, 14, and 13 cycles). Bottom plots show same data plotted as theta phase firing histograms. The order was established by high CA3 cycles. Note significant phase shift of putative superficial units #21 and 34 at bottom and the deep unit #40 at top. **e** Phase shift differences (high CA3 phase – high ECIII phase) caused by tSC3-4 cycles (>ECIII) vs. tSC1-2 cycles (>CA3) in deep and superficial cells. Data from juxtacellular (4 deep and 3 superficial cells; open dots) and multisite recordings (70 deep and 26 superficial units; filled dots). Horizontal lines indicate mean ± SD from each sublayer (deep: black; superficial: gray). Circular ANOVA, $F_{(1,101)} = 14.3$, $p = 0.0003$ for all cells together; $F_{(1,5)} = 4.8$, $p = 0.0795$ for juxtacellular cells alone; ***$p < 0.001$. **f** Mean population polar plots showing phase shift effects of high ECIII inputs in deep and superficial cells separately. Same data set as in (**e**). The arrows reflect the main trends reported in (**e**).

allowed us to evaluate the concurrent phase-locking dynamics of multiple units during high CA3 vs. high ECIII (Fig. 5d).

We found different phase-locking trends of deep and superficial cells during cycles with high ECIII versus CA3 dominant classes (Fig. 5E, circular ANOVA, $F_{(1,101)} = 14.3$, $p = 0.0003$ for all cells together; $F_{(1,5)} = 4.8$, $p = 0.0795$ for juxtacellular cells alone). During high ECIII inputs, deep cells either advance or delay their firing but did not shift as a population as compared to high CA3 cycles (Fig. 5f; mean at $-35.8 \pm 27.8°$). In contrast, most superficial cells tended to advance their firing during high ECIII vs. high CA3 inputs ($49.8 \pm 63.2°$, different than 0°, $t_{(28)} = 2.5$, $p = 0.019$; Fig. 5e, f). Therefore, spike timing of CA1 pyramidal cells could be biased differently by the two major glutamatergic pathways.

## Discussion

Our work identifies multidimensional mechanisms underlying firing selectivity of hippocampal pyramidal cells during theta

oscillations. Several intrinsic and microcircuit factors are critical to shape integration of a collection of theta-modulated inputs in individual pyramidal cells. Dissecting how each of these factors determines firing behavior in vivo is experimentally intractable given the lack of complementary tools to causally interfere with some of them at once. We thus exploited realistic computational models and evolutionary algorithms to treat this problem deterministically using heterogeneous synthetic cells in terms of morphology, biophysical and connectivity properties. This unbiased approach was combined with experimental data to dissect the mechanisms underlying the internal organization of theta phase-locking behavior in the dorsal CA1. Altogether, our data identify a set of testable physiological mechanisms underlying cell-type specific phase-locked firing. We propose that these basic mechanisms act to set preferred theta phases differently in deep and superficial CA1 pyramidal cells, while preserving individual variability, consistent with experimental data[26,45]. These internal firing reservoirs of preferred theta phases can be

instrumental for flexible representations of fluctuating relevant inputs during cognitive tasks.

We found that perisomatic inhibition delivered by complementary populations of PV and CCK basket cells represents a major factor in setting phase preference. In CA1, PV basket cells innervate preferentially pyramidal cells at the deep sublayers while CCK basket cells are more likely to target superficial pyramidal cells[36,37]. Thus, different connectivity by the two basket cell populations was enough to bias deep and superficial pyramidal cells in response to input pathways (Fig. 3). This makes sense given the coincident dynamics between ECIII inputs and CCK basket cell firing at the theta peak, and CA3 inputs and PV basket cell firing at the falling phase[11,12,46]. This result supports significant effects of pyramidal cell-types in explaining the bimodality observed in the population of juxtacellular recorded cells in basal conditions (Fig. 1) and is consistent with reports of different phase shifts across sublayers during behavioral tasks[24–26]. Interestingly, our simulations also predict that different innervation by bistratified and OLM interneurons will have more impact in deep than superficial pyramidal cells (Supplementary Fig. 4), suggesting additional roles in setting phase-locking preference of pyramidal neurons[47,48].

Our computer simulations identified additional mechanisms as well. First, we found that individual variability in passive and active intrinsic properties could account for different distributions of phase-locking preference (Fig. 2f). Data support complementary theta resonant effects of dendritic HCN channels together with sodium and potassium perisomatic currents[49,50]. Interestingly, single-cell transcriptomics suggest gradient distribution in the expression of ionic channels and receptors across CA1 sublayers, in particular HCN channels and mGluR1 (ref. [41]). In CA1 pyramidal cells a CB1-HCN pathway interfere with dendritic integration of glutamatergic inputs in superficial but not in deep cells[51]. Thus gradient expression of intrinsic properties can interact with phasic excitation and inhibition to shape phase-locking preference at single-cell level[52–54]. Second, we noted a cooperative effect between neuronal morphology and intrinsic properties to skew phase preference in a given cell (Fig. 2f, g). This result has earlier echoes in reports on the critical role of branching patterns in firing behavior and dendritic back-propagation[55]. More recent studies support the idea that appropriate matching between integrative properties and innervation patterns can critically determine preferential responses of a range of hippocampal cells to their input pathways[56,57]. Therefore, different dendritic branches of superficial and deep cells may additionally contribute to bias their phase preference[19,40]. Consistently, superficial cells with more apical dendritic branches are more responsive to CA3 inputs[36], while deep cells with more basal dendrites are more biased by CA2 inputs[58]. Similarly, recent data supporting preferential responses of proximal deep CA1 pyramidal cells to medial entorhinal inputs are consistent with our results[59]. Finally, we found that spike timing distribution was not necessarily correlated with the phase-locking preference of dendritic potentials (Fig. 4c). This is in agreement with nonlinear integrative properties of CA1 pyramidal cells[38,39] and suggests different physiological mechanisms underlying phase dynamics during theta oscillations.

Our logistic regression model also indicated that firing selection could be tuned distinctly across individual cells depending on the relative contribution between factors. We suggest that some of these major factors could change dynamically in vivo to determine phase timing during theta oscillations. Consistently, somatodendritic integration of fluctuating CA3 and ECIII glutamatergic inputs defined cycle-by-cycle by nested waveforms[42] demonstrated that firing selection is tunable (Fig. 5). Interestingly, depolarization of silent CA1 pyramidal cells by artificial

manipulations or sustained dendritic plateau are known factor that modulate the oscillatory dynamics of CA1 place cells[38,60,61].

Thus, the specific phase timing can shift in individual cells depending on different activity levels at input pathways associated to a range of cognitive demands such as encoding, retrieval[24,62] and changes of brain states[63,64]. Accordingly, experience-dependent plastic changes in the circuit built after experience may indeed affect phase-spiking relationships[16]. Notably, we found consistent trends of superficial cells to advance their firing along phases in response to higher fluctuations of entorhinal inputs recorded in basal conditions. This resembles basic mechanisms of theta phase precession during place field crossing[9]. Although many deep pyramidal cells behaved similarly, others more typically delayed their firing in response to higher ECIII inputs. Thus, our data suggest that at least some of the behavioral theta sequences built during place field crossing may actually occur in reverse order (Fig. 5d–f), a possibility that has been already noted experimentally but so far lacks mechanistic interpretation[65]. We predict that such reversed theta sequences may be instrumental for hippocampal codes.

In summary, we propose that sublayer gradients in the distribution of intrinsic, morphological and microcircuit factors collude to set preferred theta phase-locked firing differently across deep and superficial CA1 pyramidal cells. Such preferred theta phases represent stable attractors or firing reservoirs that dominate the internal hippocampal dynamics under basal conditions. Interaction between these internally generated reservoirs and fluctuating inputs associated to different cognitive demands is critical to build more flexible hippocampal representations[24,65–67].

## Methods

**Animals**. All protocols and procedures were performed according to the Spanish legislation (R.D. 1201/2005 and L.32/2007) and the European Communities Council Directive 2003 (2003/65/CE) for animal research. Experiments were approved by the Ethics Committee of the Instituto Cajal and the Spanish Research Council. A number of recordings in freely moving rats were obtained at the University of Szeged, Hungary, and were approved by the Animal Care Committee of the University of Szeged.

A total of 25 males and females mice were used from wild-type (C57BL/6J, $n = 15$), PV-Cre (B6;129P2-Pvalbtm1(cre)Arbr/J; Jackson Labs, $n = 9$), Calb1-Cre (Calb1-2A-dgCre-D; Jackson Labs, $n = 3$), and Thy1-ChR2-YFP (B6.Cg-Tg(Thy1-COP4/EYFP)18Gfng/J, Jackson Labs; $n = 2$) lines. We also used 28 wild-type males and females Wistar rats. Animals were maintained in a 12 h light–dark cycle (7 a.m. to 7 p.m.) with access to food and drink ad libitum.

**Head-fixed preparation**. For recordings under head-fixed conditions, mice were first implanted with fixation bars. To this purpose animals were anesthetized with isoflurane (1.5–2%) in oxygen (30%), while continuously monitored with an oximeter (MouseOx; Starr Life Sciences). Bars and ground/reference screws (over the cerebellum) were fixed with dental cement. After surgery, mice were habituated to head-fixed conditions over 2–3 weeks. The apparatus consisted in a cylindrical treadmill (40 cm diameter) equipped with a sensor to estimate speed and distance traveled analogically. The system was coupled to a water delivery pump controlled by a dedicated Arduino. Water drop delivery was coupled to a sound to reinforce associational learning. Animals learnt to run freely in the cylinder for water reward. During training sessions (2 sessions × day), their access to water was restricted to the apparatus. After a couple of weeks of training, animals were able to stay comfortable in the system for up to 2 h with periods of running, grooming, immobility and sleep. Access to water was removed during recording sessions to avoid motivational influences.

Once habituated to the apparatus, animals were anesthetized with isoflurane to perform a craniotomy for electrophysiological recordings (antero-posterior (AP): −3.9 to −6 mm from Bregma; medio-lateral (ML): 2–5 mm). The craniotomy was sealed with Kwik-Cast silicone elastomer and animals returned to their home cage. Experiments started the day after craniotomy.

**Freely moving preparation**. For recordings under freely moving conditions, rats were first implanted with dedicated microdrives for juxtacellular recordings. For the manual microdrive, animals were implanted with a plastic holder targeting a small craniotomy (4.3 mm posterior to Bregma and 2.5 mm lateral) under isoflurane anesthesia (1.5–2% mixed in oxygen 400–800 ml/min). The dura mater was left intact and craniotomy was cleaned with 0.05–0.07 mg/ml Mitomycin C (Sigma) to reduce growth-tissue, filled with agar (2.5–3.5%) and covered to avoid drying. A

ground epidural platinum/iridium wire (125 μm) was implanted over the cerebellum for reference. A contralateral intra-hippocampal tungsten wire (122 μm) was implanted for local field potential (LFP) recordings (reference screw at the occipital region). Cleaning of craniotomy was repeated over 4–5 consecutive days after surgery. Animals were habituated daily to the recording arena (40 × 40 × 23 cm or 25 × 25 × 35 cm) and microdrive-holder manipulation with water and food ad libitum. The day of recording, the microdrive housing a glass pipette (1.0 mm × 0.58 mm, ref. 601000; A-M Systems filled with 1.5–2.5% Neurobiotin in 0.5 M NaCl; impedance 8–15 MΩ) was mounted in the holder and gently advanced into the brain (350 μm per revolution at 3–5 μm resolution).

For the motorized microdrive, rats were implanted with the plastic holder carrying a mock glass pipette using similar coordinates, material and procedures as described above. The drive used a miniature brushless DC motor (125 : 1 planetary gear reduction; Microcomo, Faulhaber Group, Germany) attached to a screw (0.2 mm pitch, 1.6 mm diameter) and connected to the pipette holder. An Arduino-based controller was used to drive the motor at steps of 1, 10, 100, and 1000 μm in both directions. A contralateral intra-hippocampal tungsten was used to record LFP signals. A ground epidural platinum/iridium wire and/or chlorinated silver wires (125 μm) were implanted over the cerebellum and used as reference for the glass pipette. An occipital screw was implanted for independent ground and/or reference. Over the next 2–3 days, rats were habituated to the recording arena (25 × 25 × 35 cm) and manipulation of the implant. A video camera was used to monitor animals' behavior (sleep, running, grooming, rearing) and to track their position in the recording arena. The day of recording, the animal was briefly sedated with isoflurane (1.5% mixed in oxygen 400–800 ml/min) to mount a new glass pipette and returned to the home cage while monitoring LFP signals. Recordings started after 3–4 h, when rats behaved normally and LFP activity was similar than before sedation.

**Electrophysiological recordings**. For LFP recordings in head-fixed conditions, we used 16- and 32-channel silicon probes from Neuronexus (linear arrays with 50–100 μm resolution and 413–703 μm$^2$ electrode area; Poly3 arrays with 25 μm resolution and 177 μm$^2$ area and Buzsaki probes). LFP recordings in freely moving conditions were obtained from a single tungsten wire. Wideband (1 Hz–5 KHz) LFP signals were pre-amplified (4–10 × gain) and recorded with different AC multichannel amplifiers (Multichannel Systems or the RHD2000 Intan USB Board running under Open ephys). Characteristic features such as the laminar profile of theta and sharp-wave ripples, as well as unit activity were used to inform laminar position within the hippocampus. In all cases, electrode position was histologically confirmed[68].

Single-cell recordings followed by juxtacellular labeling for post-hoc immunehistochemical identification were obtained in combination with LFP. For juxtacellular recordings, a glass pipette (1.0 mm × 0.58 mm, ref. 601000; A-M Systems) was filled with 1.5–2.5% Neurobiotin in 0.5 M NaCl (impedance 8–15 MΩ). Juxtacellular signals were acquired with an intracellular amplifier (Axoclamp 2B; Axon Instruments) at 100× gain. Single-cell and simultaneous LFP recordings were sampled at 20 kHz/channel with 12 bits precision (Digidata 1440; Molecular Devices). After recording, cells were modulated using the juxtacellular labeling technique with positive current pulses (500–600 ms on–off pulses; 5–18 nA)[68].

**Chemogenetic experiments**. PV-Cre mice were injected with AAV8-DIO-hSyn-hM4D(Gi)-mCherry (1 μL; titer 5.3 × 10$^{12}$ vg/ml; provided by UNC Vector core, Roth lab) targeting the dorsal CA1 region (−1.9 mm AP; 1.25 mm ML and 1 mm depth). After a minimum of 3 weeks from injection, animals were operated for implanting head-fixing bars as described above. For juxtacellular experiments, CNO was injected once the pipette was over the hippocampus and we started searching for cells. Most cells were recorded between 10 and 60 min after CNO injection. We monitored continuously the CNO effect using parallel LFP recordings. For multisite experiments, CNO was administered after having recorded more than 20 min baseline activity. To facilitate longitudinal recordings from the same population of cells, we implanted an intraperitoneal cannula for administering CNO (5 mg/kg in dimethyl sulfoxide (DMSO)) or vehicle (DMSO) in a group of mice. We also examined off-target effects of CNO in a group of wild-type mice.

Specificity of viral expression (as in Fig. 3c) and localization of probe tracks were always assessed after experiments to validate local effects.

**Optogenetic experiments**. Calb1-Cre mice were injected with AAV5-DIO-EF1a-hChR2-EYFP (1 μL; titer 4.5 × 10$^{12}$ vg/ml; provided by UNC Vector core, Deisseroth lab) targeting the dorsal CA1 region (−1.9 mm AP; 1.25 mm ML and 1 mm depth). Two days after AAV injection, two doses of trimethoprim (TMP T0667 from Sigma; ~0.17 g/kg from saturated solution of 14 mg TMP in 1 ml saline) per day were intraperitoneally injected over 3 consecutive days to induce Cre-mediated recombination. After a minimum of 3 weeks from injection, animals were operated for implanting head-fixing bars as above. Transgenic Thy1-ChR2-YFP mice were directly implanted with fixation bars. One week after recovery from surgical implantation, animals were trained in the head-fixed apparatus as above.

Recording and optogenetic stimulation were performed with integrated micro-LED optoelectrodes originally provided by Euisik Yoon under the NSF-funded NeuroNex project and later purchased from NeuroLight Technologies, LLC,

N1-A0-036/18. To constrain precisely optical stimulation at individual deep and superficial pyramidal cells, and to avoid confounding microcircuit effects, we exploited high-density recordings and nanowatt blue light stimulation at 10–20 nW to activate isolated units[29]. The deep or superficial location of opto-tagged units was confirmed by using information about the sharp-wave ripple profile across high-density optoelectrodes. Only unambiguous deep and superficial pyramidal-like opto-tagged units were included in the analysis.

Specificity of viral expression (as in Fig. 1h) and localization of probe tracks were always assessed after experiments to validate local effects.

**Tissue processing and inmunohistochemistry**. After experiments, animals were perfused with 4% paraformaldehyde and 15% saturated picric acid in 0.1 M, pH 7.4 phosphate-buffered saline (PBS). Brains were postfixed overnight, washed in PBS, and serially cut in 70 μm coronal sections (Leica VT 1000S vibratome). Sections containing the stimulus and probe tracks were identified with a stereomicroscope (S8APO, Leica). Sections containing Neurobiotin-labeled cells were identified by incubation in 1 : 400 Alexa Fluor488-conjugated streptavidin (Jackson Immu-noResearch 016-540-084) with 0.5% Triton X-100 in PBS (PBS-Tx) for 2 h at room temperature (RT). Slices recorded in vitro containing Alexa568 filled cells were fixed for 30 min, washed in PBS, and processed similarly to others.

Sections containing the somata of recorded cells were treated with Triton 0.5% and 10% fetal bovine serum (FBS) in PBS. After washing, they were incubated overnight at RT with the primary antibody solution containing some of the following antibodies with 1% FBS in PBS-Tx: rabbit anti-calbindin (1 : 1000, CB D-28k, Swant CB-38, RRID:AB_10000340), mouse anti-calbindin (1 : 1000, CB D-28k, Swant 300, RRID:AB_10000347), rabbit anti-somatostatin (1 : 1000, Peninsula T4103, RRID:AB_518614), rabbit anti-PV (1 : 1000, Swant AB_2631173, RRID: AB_2631173), mouse anti-PV (1 : 1000, Swant, 235, RRID:AB_10000343) and rabbit anti-NPY (1 : 1000; Peninsula, T4070, RRID: AB_518504). After three washes in PBS-Tx, sections were incubated for 2 h at RT with secondary antibodies: goat anti-rabbit Alexa Fluor633 (1 : 500, Invitrogen, A21070), goat anti-mouse Alexa Fluor488 (Jackson Immunoresearch 115-545-003) or goat anti-mouse Rhodamine Red (1 : 200, Jackson ImmunoResearch, 115-295-003) in PBS-Tx-1% FBS. Following 10 min incubation with bisbenzimide H33258 (1 : 10,000 in PBS, Sigma, B2883) for labeling nuclei, sections were washed and mounted on glass slides in Mowiol (17% polyvinyl alcohol 4–88, 33% glycerin and 2% thimerosal in PBS).

Multichannel fluorescence stacks were acquired with a confocal microscope (Leica SP5; LAS AF software v2.6.0). Morphological analyses were blind to electrophysiology. Cells were classified deep or superficial depending on their position within the Calbindin+ sublayer. The distance from the cell soma to the border between the SP and radiatum (taken at 0) was measured from confocal images using information from Calbindin and bisbenzimide staining and the ImageJ software (v 1.52a NIH Image).

To evaluate appropriate expression of hM4D(Gi)-mCherry in PV basket cells, we performed triple staining to identify mCherry+ cell types based on specific markers: PV basket cells or axo-axonic cells located at the pyramidal layer and expressing PV exclusively; bistratified cells located at the pyramidal layer and co-expressing PV and NPY or PV and SST; OLM cells located at the oriens and co-expressing PV with SST but not NPY; Ivy cells expressing NPY only. Specificity was evaluated by counting the number of PV+ cells expressing mCherry. Off-target expression was assessed by counting the number of mCherry+ cells not expressing PV (e.g. some Ivy cells). Similarly, specificity and off-target expression of ChR2-YFP in Calb1-Cre and Thy1-ChR2 mice was evaluated by immunostaining against Calbindin. Data were expressed in percentage of the total.

**Computational model**. To understand how deep and superficial CA1 pyramidal cells integrate theta-modulated glutamatergic and GABAergic inputs, we modeled single cells using the Neuron (v.7.4)+ Python (v.7) platform running on an Intel Xeon E3 v5 processor with 64GB RAM and Ubuntu (v.16.04). Our simulations are based on a realistic full-scale model of the CA1 microcircuit able to autonomously generate theta oscillations[33]. To reduce computational cost and given our focus in the integrative properties of single neurons, we considered only multi-compartmental pyramidal cells with realistic morphologies and a collection of theta-modulated glutamatergic and GABAergic inputs based on the full model and literature (see Supplementary Tables 2 and 3). As the full model was conceived to simulate the CA1 microcircuit, pyramidal cells were simplified to a few numbers of compartments and basic biophysical features (see below). Given our focus on the integrative properties of single pyramidal cells, we chose to sophisticate the model details by including a range of ionic currents with a more realistic somatodendritic distribution based on well-validated previous models of CA1 (refs. [69,70]) (Supplementary Tables 1 and 2).

To consider cellular heterogeneity, we included four different realistic morphologies of CA1 pyramidal cells. They were obtained from http://neuromorpho.org/ (morphologies n128, n127 and n409 from the Turner archive and morphology sup1 corresponding to the superficial cell I040913C7SEC8_2D from the Prida archive). Each morphology was fitted to 200–300 compartments equipped with a set of active and passive voltage-dependent conductances (320 for morphology n128, 261 for sup1, 270 for n409, and 230 for n127). Each compartment was modeled using the Hodgkin–Huxley formalism with a set of

differential equations representing the dynamics of different ionic channel plus the intracellular calcium concentration (18 equations per compartment). Briefly, the total current flowing through each compartment was the result of different contributions:

$$I = \sum I_{\text{syn}} + \sum I_{\text{intrinsicactive}} + \sum I_{\text{intrinsicpassive}} \qquad (1)$$

Each passive, active or synaptic current was generically modeled following the HH formalism as:

$$I = g(V, t) \cdot (V - V_{\text{rev}}) \qquad (2)$$

With specific HH dynamics as specified in Supplementary Tables 2 and 3 (see model DB codes).

To fit free parameters, we adopted a combination of GAs[34] and random selection within experimental value ranges (Supplementary Table 1). For GA fitting, the effective conductance of each channel was modeled as the product of the maximal experimental conductance (Gmax) and a GA factor:

$$g = G_{\text{max}} \cdot GA_{\text{factor}} \qquad (3)$$

The intrinsic GA consisted on 100 generation runs to look for parameter values that converge to the target behavior. During one generation, each model was assessed to target experimental behavior (Fig. 2b, c). Successful GA factor values were passed with a 40% probability to next generations as well as 20% probability for unsuccessful values to promote variability. To avoid GA factors to get trapped, 40% of values were "mutated": one GA factor at once was changed randomly within the range.

Each compartment was passively connected to neighbor compartments by axial resistance. Passive properties were the following: (a) internal resistivity was defined as a Ra = 100 (Ohm × cm) scaled by a GA factor; (b) membrane capacitance at soma was 5 μF/cm², and 1.8 × 5 × SpineFactor (μF/cm²) for the rest of the compartments. The SpineFactor (SS) accounts for the extra surface area of dendritic spines in a given compartment (Supplementary Table 2)[71]. Specifically, SS = 2.51 for basal dendrites. For apical dendrites located <350 μm from soma: SS = 1.69 if thickness > 1.6, SS = 1.60 if 0.55 < thickness < 1.6, SS = 1.86, if thickness < 0.55, and for apical dendrites > 350 μm from soma, SS = 2.10 if thickness > 0.35, SS = 1.71 if thickness < 0.35. Finally, the specific membrane resistivity (S/cm²) was defined following:

$$g_{\text{pas}}^{\text{soma}} = \frac{0.001}{80 + \frac{0.4 - 80}{1 + \exp((225 - z)/30)}} \qquad (4)$$

$$g_{\text{pas}}^{\text{rest}} = SS \cdot g_{\text{pas}}^{\text{soma}} \qquad (5)$$

where z is the somatodendritic distance.

Different glutamatergic (CA3, CA2, ECIII, and ECII) and GABAergic (Axo, Bis, CCK, Ivy, NGF, OLM, PV, and SCA) inputs were simulated along the somatodendritic compartments following realistic distributions (Supplementary Table 3; columns # of boutons, Location and Distance to soma). For synaptic currents, individual synapses were modeled using NetCon High Order Calculator (HOC) function and distribution along the morphology followed experimental observations (Supplementary Table 3). Distribution along theta cycles followed an asymmetric Gaussian (beta function) with parameters β1 and β2 determined by experimental observations (Supplementary Table 3).

Synapse dynamics were set with the Exp2Syn HOC function, as in the full model with double exponential functions (Supplementary Table 3; columns Erev, tau1, tau2, and Gmax) representing GABAa and AMPA currents. As in the full model, NMDA receptors were not included.

$$G_{\text{max}}(t) \propto \left( e^{-t/\text{tau}_1} - e^{-t/\text{tau}_2} \right) \qquad (6)$$

We compared a subset of our simulations with those emerging from the full realistic model at scales 2 and 100 and found qualitatively similar results (Supplementary Fig. 5). The full model was run in the supercomputers Comet and Stampede of the Neuroscience Gateway portal (https://www.nsgportal.org/)[72].

To model several individual CA1 pyramidal cells, we exploited GA variability of the intrinsic and synaptic sets of values (Supplementary Fig. 3A, B). GA factors for ionic channel conductance were chosen to fit the neuron intrinsic properties[34–37]. GA factors of synaptic conductance of the CA3 inputs and their associated feedforward inhibitory inputs (Axo, Bis, CCK, PV, and SCA) were selected to target experimental data[36]. Each GA run provided a set of individuals (combination of GA factors) fitting experimental values for intrinsic and synaptic properties. We then chose 20 individuals fitting intrinsic and synaptic traits in the four different morphologies (Supplementary Fig. 3B).

To evaluate the integrative properties of individual neurons during theta oscillations, we simulated realistic theta-modulated firing of glutamatergic and GABAergic inputs impinging at each compartment (Supplementary Table 3; column Frequency and Phase; 1000 cycles per simulation and cell). Phases of different inputs were established using a sinusoidal theta signal as a reference. This signal was subsequently used to evaluate phase-locked behavior of simulated cells. The firing rate from each individual pyramidal cell was evaluated and constrained to realistic values (0.01–8 Hz). Parameter values were derived from experimental ranges reported in literature and detailed in Supplementary Table 3 (refs. [73–79]).

Model results not fitted by GA were compared with experimental values[36,51,80,81] (Supplementary Table 4).

Our model is available at Github (https://github.com/acnavasolive/LCN-HippoModel) and ModelDB: 258854 (http://modeldb.yale.edu/258854).

**Computer simulation experiments.** A major goal of our simulations was to evaluate the effect of different connectivity of pyramidal cells by local GABAergic interneurons (Fig. 3). Thus, we tested the effect of changing the full model connectivity of PV and CCK basket cells based on experimental data suggesting up to 70% difference on the number of boutons impinging onto deep and superficial CA1 pyramidal cells[36]. To this purpose, we changed the distribution of perisomatic boutons (−50 to 150 μm around the soma) to the % of the GA-fitted value to simulate deep (30% CCK, 100% PV) and superficial cells (100% CCK, 30% PV) (Fig. 3a, b and Supplementary Fig. 4A, B). We also aimed to evaluate the effect of reduction of 70% in the number of boutons for the other interneuronal types (Supplementary Fig. 4C, D). Given the lack of experimental data on this matter, we chose to simulate a general reduction all along their somatodendritic targets.

In a subset of simulations we confirmed that reducing connectivity (boutons) or the corresponding input rate (Supplementary Table 2, Frequency) yielded qualitatively similar results. To make testable predictions we simulated full reduction of PV basket cell input rates (Supplementary Fig. 6A) and proportional reduction of PV, Bis, OLM and Ivy inputs according to experimental data (Fig. 3c–e). We also tested the effect of changes of glutamatergic inputs (ECIII, CA3 and CA2) by increasing/decreasing their input rates in a realistic range (Fig. 4a).

**Logistic regression model.** For the multinomial logistic regression analysis, preferred phases from all simulations (n = 731 synthetic cells) were pooled into a big data set including detailed information from morphological (number of dendritic branches and axial resistance), intrinsic (maximal conductances of all ionic channels fitted by GA) and synaptic features (same for all synaptic conductances) (97 features in total, see full list below). The logistic regression function from sklearn Python library was used to multinomially fit data using 30° binned preferred phases as the categorical dependent variable and each features as independent variables (Supplementary Fig. 7A). The solver used was a quasi-Newton optimization method named L-BFGS.

Outcomes of the model (coefficients) indicate the effect of increasing (upregulating; black) or decreasing (downregulating; gray) a given feature on specific preferred phases (Supplementary Fig. 7B; only significant features are shown). To evaluate significant contribution of each feature in determining phase preference, we run a circular Rayleigh statistical test (p < 0.001). The weighted contribution of significant features to a given phase was considered to build a generalized linear model. Individual features (up- and downregulated) explaining at least 20% variance of phase-locking firing for three consecutive 30° bins were considered to contribute significantly and analyzed separately (Fig. 4d).

Full list of features: CA3 factor (CA3_F), original gCA3 (CA3_G), final gCA3 (CA3_F*G), CA3 #boutons (CA3_B), CA3 frequency (CA3_Freq), CA3 phase (CA3_Ph), CA2 factor (CA2_F), original gCA2 (CA2_G), final gCA2 (CA2_F*G), CA2 #boutons (CA2_B), CA2 frequency (CA2_Freq), CA2 phase (CA2_Ph), EC3 factor (EC3_F), original gEC3 (EC3_G), final gEC3 (EC3_F*G), EC3 #boutons (EC3_B), EC3 frequency (EC3_Freq), EC3 phase (EC3_Ph), EC2 factor (EC2_F), original gEC2 (EC2_G), final gEC2 (EC2_F*G), EC2 #boutons (EC2_B), EC2 frequency (EC2_Freq), EC2 phase (EC2_Ph), Axo factor (Axo_F), original gAxo (Axo_G), final gAxo (Axo_F*G), Axo #boutons (Axo_B), Axo frequency (Axo_Freq), Axo phase (Axo_Ph), Bis factor (Bis_F), original gBis (Bis_G), final gBis (Bis_F*G), Bis #boutons (Bis_B), Bis frequency (Bis_Freq), Bis phase (Bis_Ph), CCK factor (CCK_F), original gCCK (CCK_G), final gCCK (CCK_F*G), CCK #boutons (CCK_B), CCK frequency (CCK_Freq), CCK phase (CCK_Ph), Ivy factor (Ivy_F), original gIvy (Ivy_G), final gIvy (Ivy_F*G), Ivy #boutons (Ivy_B), Ivy frequency (Ivy_Freq), Ivy phase (Ivy_Ph), NGF factor (NGF_F), original gNGF (NGF_G), final gNGF (NGF_F*G), NGF #boutons (NGF_B), NGF frequency (NGF_Freq), NGF phase (NGF_Ph), OLM factor (OL-M_F), original gOLM (OL-M_G), final gOLM (OL-M_F*G), OLM #boutons (OL-M_B), OLM frequency (OL-M_Freq), OLM phase (OL-M_Ph), PV factor (PV_F), original gPV (PV_G), final gPV (PV_F*G), PV #boutons (PV_B), PV frequency (PV_Freq), PV phase (PV_Ph), SCA factor (SCA_F), original gSCA (SCA_G), final gSCA (SCA_F*G), SCA #boutons (SCA_B), SCA frequency (SCA_Freq), SCA phase (SCA_Ph), iNa factor (iNa_F), gNa original mean (iNa_Gmean), iA factor (iA_F), gA original mean (iA_Gmean), iAHPs factor (iAHPs_F), gAHPs original mean (iAHPs_Gmean), iC factor (iC_F), gC original mean (iC_Gmean), iKDR factor (iKDR_F), gKDR original mean (iKDR_Gmean), iM factor (iM_F), gM original mean (iM_Gmean), iCa factor (iCa_F), gCa original mean (iCa_Gmean), HCN factor (HCN_F), gHCN original mean (HCN_Gmean), L factor (L_F), gL original mean (L_Gmean), Ra factor (Ra_F), gRa original mean (Ra_Gmean), #Apic branches (nBrApic), #Basal branches (nBrBasal), and #Total branches (nBrTotal).

**Analysis of experimental LFP signals.** All analysis was performed using routines written in Matlab 9.5 2018b (MathWorks). For experimental data, LFPs from different layers were identified according to distinctive features, including sharp-wave ripples (at stratum radiatum (SR) and stratum pyramidale (SP)) and maximal

theta oscillations (stratum lacunosum moleculare (SLM)). In data from juxtacellular recordings, we used the LFP signal from the glass pipette similarly. Power spectra of LFP signals were estimated using the Fast Fourier transform. For theta activity, non-overlapping segments of continuous oscillations in the 4–12 Hz band were identified using the largest amplitude channel (typically at SLM). To detect theta cycles, we band-filtered LFP signals at 4–12 Hz, with forward-backward-zero-phase FIR filters, to detect troughs and validate them whenever they were surrounding by equivalent peaks at the expected period defined by the oscillatory spectrum. We next identified the associated theta peak at the SP, which was used as a reference. For sharp-wave ripples, LFP signals at SR were low-pass filtered (<100 Hz) to identify sharp waves and signals from SP were bandpass filtered (100–600 Hz) to identify ripples. For detecting sharp waves, filtered signals were smoothed (Gaussian kernel) and events detected by thresholding >3 SDs. For detecting ripples, bandpass-filtered signals were smoothed (Savitzky–Golay) and events detected by thresholding >2 SDs. All pairs of detected events were visually confirmed and artifact discarded.

**Unsupervised decomposition of theta cycles**. Based on recent results supporting distinctive nested spectral components[25,42], we applied unsupervised SOM analysis to classify theta cycles according to their LFP signatures[43]. To this purpose, each cycle was mapped into a high-dimensional space determined by downsampling with principal component analysis to explain >90% variance. Cycles sharing similar nested components (oscillations in the beta, low gamma and high gamma band) will map together in the hyper-space. SOM works to organize this cluster of points in the hyper-space into a low-dimensional matrix by optimizing the topological distance. Cycles grouped in a given element of the matrix share common waveforms and neighboring elements that are likely to be similar were grouped together in a similarity matrix. The number of clusters was defined by the Davies–Bouldin index. SOM represents a predictive model that needs to be validated. Thus, for each mean element of the matrix we represented the time-frequency components using wavelet analysis and used information from the similarity matrix to identify theta-nested spectral components tSC1 (<20 Hz centered at the theta peak), tSC2 (30–40 Hz, at the falling phase), tSC3 (50–60 Hz; at the theta peak), and tSC4 (70–90 Hz at the peak), as validated recently[42]. The SOM classifier (RhythSOM) is available at GitHub (https://github.com/acnavasolive/RhythSOM).

**Analysis of juxtacellular recordings**. For juxtacellularly labeled cells, signals from glass pipettes were high-pass filtered at 300 Hz to detect positive spikes from the recorded cell (>8 SD). Simultaneous LFP signals at SLM and SP were processed as described above. The stability of the action potential waveform (peak-to-peak duration and amplitude as well as a spike asymmetry index defined as the ratio of the difference between the negative and positive baseline-to-peak amplitudes and their sum) was evaluated over the entire recording session (>3 min), before juxtacellular electroporation. Interspike interval autocorrelograms (1 ms bin size) were constructed using all detected spikes. Baseline firing rate was stable for small movements of the pipette towards the cell, excluding mechanical interferences.

Phase-locked firing of single cells was evaluated from each spike using the Hilbert phase of theta peaks detected before (either all detected cycles or classified cycles tSCi). Each theta cycle was divided into 25 bins. Phase locking was quantified using the mean vector length of phase distribution from 0 to 1 in significantly theta-modulated cells ($p < 0.05$, Rayleigh test). The SP theta trough was set at 0 (360°) and peaks at 180°. To test phase-locking significance, we implemented a surrogate test consisting of randomizing 1000 times the firing rate of each cell across all cycles. Differences between the experimental/simulated and surrogate distribution were tested at $p = 0.05$ significant level.

**State-dependent analysis of juxtacellular recordings**. To account for state-dependent effects on single-cell firing, we defined periods of sleep, immobility and other activities (grooming, whisking, and arousal) using an integrated analysis of behavioral and electrophysiological data.

In the open-field preparation, sleeping epochs were defined as periods of immobility, lasting at least 60 s, and typically associated with curled-up postures as evaluated from the video. They were typically associated with slow-wave high-amplitude activity (<3 Hz) and short REM phases characterized by low amplitude theta activity. Running periods were defined whenever the rat was moving at >5 cm/s and theta oscillations were recorded in the hippocampus. Immobility was defined for >5 s periods at 0 cm/s with large irregular activity. Wake non-running periods were defined as periods of immobility associated with hippocampal theta activity and included behaviors like grooming, attention, whisking, or head nodding.

In the head-fixed preparation, sleep was defined whenever mice were immobile with eyes closed and a higher arousal level (not responding to mild clapping). Sleep in the wheel was accompanied by high-amplitude slow-wave activity and occasionally by theta. Only a minority of mice exhibited REM sleep in the wheel and therefore this state was not considered for analysis. Running was defined whenever the speed was >4 cm/s in association with theta oscillations. Immobility periods were defined as continuous periods (>5 s) at 0 cm/s in the wheel accompanied by large irregular activity. During some immobility periods, mice performed grooming, whisking, and attentive behaviors that were associated with hippocampal theta oscillations.

**Sorting and analysis of individual units from multisite recordings**. Single units were isolated using Kilosort2 (https://github.com/MouseLand/Kilosort2)[82]. Spike trains were analyzed by generating interval time histograms and temporal autocorrelograms (±0.025 s). Only units with >100 spikes, none of them in the refractory period of the interspike time histogram (1–2 ms), and with spike amplitudes three to four times above background noise, typically 20–30 μV, were included. Putative pyramidal cells and interneurons were differentiated following standard criteria, including trough-to-peak duration of the nonfiltered spike, waveform asymmetry, and the first moment of the autocorrelograms. The firing rate was not used as a classificatory criterion, given the large variability, especially between interneuronal types and states. Putative interneurons were subclassified (PV, CCK, OLM, and Bis) according to their theta preference and behavior during sharp-wave ripples, as previously reported[83]. Deep and superficial pyramidal cells were subclassified according to their location in the multisite probe (maximal amplitude spike) and the associated sharp-wave ripple waveform in the corresponding channel, as reported before[84]. Units that could not be unambiguously identified were left unclassified.

Rhythmic and phase-locked firing of sorted units was evaluated as described above for juxtacellular data.

**Rank order test**. To identify theta cycles with simultaneous unit firing more than expected by chance, we used a rank order test[44]. For each cycle, the timing from multiple pyramidal units recorded simultaneously with multisite probes were transformed in a normalized chronological order from 0 to 1, to avoid influences from instantaneous variations of theta frequency. Only cycles with at least four participating units were included. Only the first spike from each unit in a cycle was considered for the rank order. We then constructed 500 rank order shuffles for each cycle and tested each real sequence in a cycle against all shuffle distributions using Pearson's correlations. Rank distribution of the correlation values was tested against shuffle correlations; significance at $p = 0.05$. Only cycles with more significant correlations than expected by chance were considered for further analysis.

**Analysis of simulation results**. Results from simulations were analyzed with Matlab routines. Action potential firing was estimated from the high-pass filtered (>300 HZ) somatic membrane potential signal by thresholding. Membrane potential fluctuations from the soma, the proximal apical and the distal main dendritic compartments were used to evaluate somadendritic dynamics. To evaluate phase-locked behavior of simulated cells the sinusoidal theta signal was used as a reference and analysis implemented as previously described for juxtacellular recordings.

**Statistical analysis**. Statistical analysis was performed with Matlab. No statistical method was used to predetermine sample sizes. Normality and homoscedasticity were evaluated with the Kolmogorov–Smirnov and Levene's tests, respectively. The exact number of replications for each experiment is detailed in text and figures.

Circular statistics was performed using the CircStat Matlab toolbox. Circular one-way (Watson–Williams multi-sample test) or two-way ANOVAs (Harrison–Kanji tests) were applied to examine effects of preparation and/or sublayers. Post-hoc comparisons were evaluated with T-tests, corrected by Bonferroni whenever requiered. Correlations were evaluated with the Pearson product-moment correlation coefficient, which was tested against 0 (i.e., no correlation was the null hypothesis) at $p < 0.05$ (two sided). Both the Pearson's coefficient and $p$-value are reported to facilitate interpretation.

**Reporting summary**. Further information on research design is available in the Nature Research Reporting Summary linked to this article.

## Data availability
Experimental data supporting the findings of this study are available on request from the corresponding author. The model and validated parametric sets are available at Github https://github.com/acnavasolive/LCN-HippoModel and ModelDB: 258854. The Matlab SOM routine adapted to the analysis of LFP events (RhythSOM) is available at GitHub (https://github.com/acnavasolive/RhythSOM).

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

## Acknowledgements

This study was supported by grants from the Spanish Ministerio de Economía y Competitividad (BFU2015-66887-R and RTI2018-098581-B-I00) to L.M.P. R.G.A. acknowledges support by the Hungarian National Office for Research and Technology GINOP-2.3.2-15-2016-00018 to Gábor Tamás. We thank Euisik Yoon and John Seymour for introducing us to the integrated micro-LED optoelectrode technology. The micro-LED optoelectrodes were initially provided by the NSF-funded NeuroNex Hub MINT (Multimodal Integrated Neural Technologies) at the University of Michigan under NSF 1707316. A.C.N.O. is supported by an PhD Fellowship from the Spanish Ministry of Education (FPU17/03268). We thank Pablo Varona, Jose Manuel Ibarz, and Thomas Hainmueller for useful comments and suggestions. This work used the supercomputer clusters provided by the Neuroscience Gateway portal (https://www.nsgportal.org/). Thanks to Ivan Raikov for guiding us with computational simulations of the full Soltesz model in the supercomputer clusters. We thank Enrique R Sebastian for helping us with statistical validation of the RhythSOM classifier.

## Author contributions

L.M.P. designed the study. A.N.O. ran computer simulations. M.V., A.S., T.J.P., R.G.A., G.G., and E.C. obtained data. A.N.O., M.V., A.S., and T.J.P. analyzed data. L.M.P. wrote the paper.

## Competing interests

The authors declare no competing interests.
