## [Peer Review File · Nature Communications]

Reviewers' Comments:

Reviewer #1:

Remarks to the Author:

This study investigates cellular and synaptic mechanisms that control the phase preference of CA1 pyramidal cells during theta activity. Experimental evidence is presented that superficial and deep pyramidal neurons differ in their phase preference. A combination of sophisticated modelling approaches and experiments are then used to suggest roles for neuronal morphology, ion channel expression, strength of input from CCK and PV interneuron populations, and relative activity in EC and CA3 inputs. The major message appears to be that all of these mechanisms interact to affect theta phase locking.

The possible importance of the study is that it illustrates how in vivo neuronal activity can be influenced by interactions between cellular and circuit level properties. This is perhaps not surprising, but the modelling approach is nice in that it quantifies how different types of properties interact with one another. The authors make a good point in that whereas individual properties, for example a class of ion channel or an input from a particular interneuron class, are usually investigated in isolation, the results presented here suggest the importance of interactions between heterogeneous properties. Some general concerns are that some of the effects don't appear all that convincing and the extent to which the models selected for analysis correspond with real neurons isn't really clear.

Major

1. It's not clear how well the models represent real neurons. I appreciate the approach of evaluating and selecting from multiple models using a genetic algorithm, but it's not clear how well the space of possible models has been evaluated, or whether the targets for fitting are sufficient to guarantee a model that accounts well for other properties of CA1 pyramidal neurons. Details about the number of possible models and the number tested would be helpful. It might be helpful to also validate the models against relevant independent measures that were not used for fitting. In addition, it's unclear why 21 'individuals' were randomly selected.
2. Figure 2F-H assess the impact of morphology on theta phase of firing. One of the conclusions is that "morphologies n409 and n127 show a trend for cells with deep-like traits versus n128 and sup1". Was this assessed in any statistical way to determine whether the apparent differences might have arisen by chance? From the example figure it looks like n409 is active at the trough and not the peak. Have the authors considered using morphologies from known superficial and deep pyramidal cells? This could make a much more convincing argument for the importance of the morphology of each cell type.
3. Figure 2G-H. Do the tests of significance account for multiple comparisons being made? It seems they should. If they don't then what justification is there for this?
4. Why are firing phases for the example cells in Figure 3A (all inputs) different to the phases for the same cells in 2F. The effects of morphology that were suggested for 2F-H don't obviously appear to hold up for the 'All inputs' group in Figure 3A.
5. The DREADD experiment presented in Figure 4E shows neuronal activity in the presence of CNO, but I couldn't find an analysis to show that CNO has any effect on the neurons relative to control conditions. There are some example histograms in Figure S6 but they are not accompanied by any further analysis. If there is no statistically robust effect of CNO then the results are hard to interpret.
6. Overall the manuscript is very dense, uses a lot of jargon and is often hard to follow. As just one example, a "firing reservoir" sounds impressive, but I don't think I have any idea what this is or how it would be "repurposed". The figure legends are also hard to follow. As an example, Figure 4H begins "Experimental data confirmed simulation prediction". It would be helpful to refer to the prediction (Figure

4B?) and to explain why the experimental data meets the prediction (it's not clear that it does) and if any tests were carried out to show this is not a chance result.

Minor

Line 92-93, please state the test used for the multivariate analysis. What are the 'statistical effects for sublayer'? I couldn't find details of the analysis for this data in the methods.

Line 95, "superficial cells tended to fire preferentially..." to what extent is this observation statistically robust?

Line 97 "concurrent" -> "were observed"?

Figure 1F. What is the meaning of the open and closed symbols?

Is Figure 1H necessary? I'm not sure what it adds that's useful.

Reviewer #2:

Remarks to the Author:

The authors combined electrophysiological and computational tools to investigate phase-locking of pyramidal cells in CA1. They found experimentally that superficial cells tended to fire preferentially at the theta trough and deep cells at the theta peak. Then they focused on modeling to better understand what determines this effect. They found effects of dendritic properties, but also strong effects of perisomatic inhibition. To confirm predictions of the simulation they conducted more experiments and manipulated GABAergic population. They found that the GABAergic population can indeed constraint phase selection. They also modeled the influence of CA3 and ECIII inputs using a large regression model and concluded that "internal oscillatory dynamics of CA1 pyramidal cells is determined by intrinsic and microcircuit factors that cooperate to segregate firing across sublayers".

Overall, this study combines experimental and modeling work in a very advanced way. The model is rather sophisticated and builds on top of previous modeling studies using biophysically realistic parameter values. The way the experimental part and the model complement one another is quite remarkable.

My main concerns are in the experimental aspect, which is essential for the rest of the study. The evidence for the main result that superficial cells tended to fire preferentially at the theta trough and deep cells at the theta peak comes from a relatively small number of cells (12 from mice and 28 from rats). It wasn't clear to me if each subpopulation of cells was statistically significant on its own or they were only significant when taken together. From figure 1G that shows phase preference for each cell, the only strong regularity seems to be that mouse superficial layer cells fired at trough. Is this phenomenon driving the entire effect?

The significance of the findings could be justified more. It is not surprising that different inhibitory and excitatory inputs can change the phase preference of a downstream cell. The work the authors did in identifying some mechanisms that have particularly prominent effect on this is certainly important. However, the significance of these results could be amplified if they could connect these mechanisms with circuit models of hippocampal processing. At the moment, it is not clear what could be a functional significance of their results.

The phase is reported relative to theta oscillations measured at the stratum pyramidale. What is the difference in the phase of theta for deep and superficial cells local theta and does this difference have impact on the model?

Minor comments:

It is not visible what are the 97 factors that were used in regression shown in figure S7A.

Figure 1G is very important, but it was little hard to read (circles are largely overlapped, making it difficult to evaluate the strength of the effect). Perhaps the authors could visualize this important results with an additional figure to make it more accessible.

How many cells did you analyze in the first section of the results, 12 from mice and 28 from rats or 10 from mice and 20 from rats (line 83 vs line 94)?

Reviewer #3:

Remarks to the Author:

In the current manuscript, Navas-Olive et al. use in vivo recordings to evaluate the precise relationship between the phase of hippocampal theta and the firing rate of CA1 pyramidal cells. They observe that most individual neurons have a unimodal relationship of firing rate to theta phase, but that the population displays a bimodal relationship, with some neurons preferentially firing at the trough of theta and others preferentially firing at the peak. The authors then use biologically realistic computational models modified by genetic algorithms to explore the underlying cellular mechanisms which may govern the phase-relationship of firing. They observe a complex interaction between morphology, inhibitory inputs, and excitatory inputs which largely explains the observed differences in phase locking. The authors demonstrate that these inputs can be dynamically altered to modify firing timing and perform some experimental validation of their computational findings.

In general, I find this to be an excellent scientific study and I am very enthusiastic about it. While it is true that bimodal firing with respect to hippocampal theta has been observed in prior studies, here the authors provide highly detailed and compelling arguments which help explain the underlying biological basis for this phenomenon.

I do have a few major and several minor issues with the manuscript that need to be addressed prior to publication.

Major Issues:

1) Many of the figures are poorly described, leaving the reader to guess what the authors are showing. The following list is NOT all-inclusive. The authors should go through each figure and ensure that every item in every panel is fully explained.

Figure 1A: There are four mini panels with only the following description in the figure legend: "Example of a deep CA1 pyramidal cell recorded and labeled juxtacellularly from a head-fixed running mouse." While an electrophysiologist might know that the upper right panel is the average waveform of the recorded neuron's action potentials and the panel below that is an autocorrelogram showing a clear refractory period, many readers will not, and this information needs to be included in the figure legend.

Figure 1B: The LFP traces are not described in the figure legend at all. Are the LFP traces filtered, or raw? Is the reader supposed to be able to see something in those raw traces? It seems like the traces are far too zoomed out to see how the spikes align with the LFP traces.

Figure 1F: What are the solid vs. open circles? What is black vs. green?

Figure 1G: The figure legend says this is multivariate analysis. This might be what accounts for the p-values in the bottom right of the panel, but it is not clear what is actually being plotted in the primary figure of that

panel. I presume that figure is the phase of peak firing for each neuron (with mean and SEM), but the authors should not make the reader guess. Also, when the multivariate analysis finds statistical effects of "preparation", does that mean there is a difference between rats and mice? If so, I encourage the authors to use the term "species" or "rat vs. mouse" rather than "preparation." If not, please explain what is meant by "preparation."

Figure 2G-H: What do the dots represent and what data are the box-and-whisker plots based on? I presume the dots are the values for each individual for the four morphologies. Was each individual only simulated once per morphology? If so, and if I am interpreting the graph properly, are the box-and-whisker plots based on only those four datapoints? If so, I'm not sure that the plots are correct. For example, individuals 12 and 15 have what appear to be three datapoints near 360 and one point near 180, but the errorbar extends all the way to ~450. If those are typical box-and-whisker plots, and they are using only the four plotted datapoints, they are not correct. If, on the other hand, the box-and-whisker plots are drawn from other data (perhaps many simulations of each individual/morphology pair), that should be made clear in the figure legend. And if the dots represent the average phase preference for each morphology across many simulations of each individual, this should also be made clear. All of these points apply to Figure 2H as well.

Figure 4E: If I am interpreting the figure correctly, I believe that "...the matrix on the right..." is actually referring to the matrix on the left-hand side of that panel.

Figure 4H: What is yellow vs. orange? I presume that is juxtacellular vs. multisite, but this should be clear.

Supplemental Figure S8: The description of panel A mentions numbers in parentheses. But I don't see any numbers in parentheses in panel A.

2) The data presented in Figures 3D and 3E needs to be quantified for statistical significance. It is true that those two graphs appear similar, but the authors should quantify whether the phase locked firing in Figure 3E is statistically more similar the prediction in Figure 3D than

1) chance (using shuffles);

and 2) the phase locked firing from control cells (such as the data from Figure 1D).

3) In addition, much of the data in Figure 4 requires statistical quantification. Are the shifts observed in panel 4E significantly different between deep-like and superficial-like? Is the shift of either group (or both groups) significantly different from chance? The same analysis should be done for the data in panel 4I. The highlighted sequences in panel 4J also need quantification. It seems to me that for all three of the highlighted examples, the authors could have just as easily drawn diagonal lines in the opposite direction and hit just as many neurons. If the authors want to say that those are "dynamic rank order changes", they need to show that those particular patterns are unlikely to have arisen by chance. Indeed, showing only three questionable examples with no population summary analysis indicates that this final claim is not well-supported in the data.

4) It is well-established that experience-dependent firing can induce both short- and long-term changes in synaptic strengths. Given that the authors see that the relative contributions of CA3 vs. ECIII inputs can impact the preferred firing phase, I would like the authors to discuss how they foresee experience-dependent plastic changes in the circuit affecting phase-spiking relationships. For example, can these findings explain the requirement for experience in theta sequence expression (Feng et al., J. Neurosci. 2015)?

Minor Issues:

5) There are many grammatical errors throughout the paper that need to be corrected. Here is a short, NOT all-inclusive list:

Abstract: Theta oscillations play *a* major role ...

Introduction, 1st paragraph: At the core of *the* declarative memory system...

Introduction, 2nd paragraph: "CA1 sublayers" should be "CA1 pyramidal sublayers".

Page 4: ... using *morphological* reconstructions...

Page 4: Hodgkin-Huxley should be Hodgkin-Huxley

Page 4: To *constrain* the parametric space...

Page 7: Dedicated GABAergic microcircuits *constrain* phase ...

Page 7: ...across individual cells we *ran* a multinomial ...

6) Figure 2 has no Panel E.

7) Identifying sleep based on curled posture and lab notes seems like it would be challenging to replicate.

The quantification of sleep needs to use a more replicable method. For example, sleep could be quantified as periods of no movement for >30 seconds.

8) For the analysis of possible sleep-based changes in preferred firing phase (Figure 1F, for example), the authors do not separate sleep into SWS vs. REM. However, REM typically accounts for only a small percentage of sleep, and according to Mizuseki et al. (Nat. Neuro., 2011), the shift in preferred phase occurs during REM. Thus, changes in phase preference from wake to sleep could be obscured if all sleep stages are combined. The authors state that head-fixed animals did not display much REM, but if the authors also have sleep recordings from freely moving animals, these sleep periods should be separated into SWS vs. REM (on the basis of delta/theta ratios) and re-analyzed.

9) In the Methods, the authors should denote how many WT vs. PV-Cre mice were used to ensure that the data is not skewed by the near-exclusive use of one or the other genotype.

10) In the Methods, the authors constrain the firing rate from each model pyramidal cell to 0.1-5Hz. I am unsure how the authors performed this constraint. Did they randomly remove or add spikes until the firing was within the appropriate bounds? Did they remove cells that did not have inherent firing rates within these bounds? This needs to be made more clear. In addition, on page 5, the authors say they chose only individuals with firing rates between 0.01-8Hz (this is repeated in the legend for Supplemental Figure S3). I'm confused as to whether the authors used 0.01-8 Hz or 0.1-5 Hz for all analyzes or whether some analyzes used one firing rate range while other analyzes used a separate range. This should be made clear. Furthermore, all analyzes should use the same firing rate range unless there is a specific scientific reason not to.

Reviewer #1 (Remarks to the Author):

This study investigates cellular and synaptic mechanisms that control the phase preference of CA1 pyramidal cells during theta activity. Experimental evidence is presented that superficial and deep pyramidal neurons differ in their phase preference. A combination of sophisticated modelling approaches and experiments are then used to suggest roles for neuronal morphology, ion channel expression, strength of input from CCK and PV interneuron populations, and relative activity in EC and CA3 inputs. The major message appears to be that all of these mechanisms interact to affect theta phase locking.

The possible importance of the study is that it illustrates how in vivo neuronal activity can be influenced by interactions between cellular and circuit level properties. This is perhaps not surprising, but the modelling approach is nice in that it quantifies how different types of properties interact with one another. The authors make a good point in that whereas individual properties, for example a class of ion channel or an input from a particular interneuron class, are usually investigated in isolation, the results presented here suggest the importance of interactions between heterogeneous properties. Some general concerns are that some of the effects don't appear all that convincing and the extent to which the models selected for analysis correspond with real neurons isn't really clear.

We appreciate the reviewer's positive view. Below we address the specific concerns.

Major

1. It's not clear how well the models represent real neurons. I appreciate the approach of evaluating and selecting from multiple models using a genetic algorithm, but it's not clear how well the space of possible models has been evaluated, or whether the targets for fitting are sufficient to guarantee a model that accounts well for other properties of CA1 pyramidal neurons. Details about the number of possible models and the number tested would be helpful. It might be helpful to also validate the models against relevant independent measures that were not used for fitting.

This very important issue was poorly explained in the first version. Please, note that our approach considers the GA validation and generalization implicitly. For validation, we fit intrinsic and synaptic properties for one morphology and then generalize to others after cross-validation (see the new Fig.S3A). Regarding generalization of GA models to non-fitted features, note that theta-modulated behavior such as realistic dendritic and somatic activity (Fig.2D) and phase-locking firing (Fig.2E), all represent independent measures contributing to validation and generalization. We have now added this relevant information in the revised version. See page 5 and Supplementary Fig.3

In addition, it's unclear why 21 'individuals' were randomly selected.

This is just a number to gain running time efficiency. We run originally for 21 based on our estimations of computing time for cross-validation and generalization across morphologies. To avoid concerns regarding this minor detail we now set this value at 20. Please, note in the new version we renamed individuals according to ranking order to simplify the narrative.

2. Figure 2F-H assess the impact of morphology on theta phase of firing. One of the conclusions is that "morphologies n409 and n127 show a trend for cells with deep-like traits versus n128 and sup1". Was this assessed in any statistical way to determine whether the apparent differences

might have arisen by chance? From the example figure it looks like n409 is active at the trough and not the peak. Have the authors considered using morphologies from known superficial and deep pyramidal cells? This could make a much more convincing argument for the importance of the morphology of each cell type.

We thank the reviewer for this suggestion. Please, note we actually assessed the morphological effect statistically by considering the total number of branches as a factor in the analysis of morphological effects (Fig.2G). Also, note that morphological factors (apical and basal number of branches) emerge as a significant feature of the logistic regression model (Fig.4D). Importantly, it is reported that superficial cells tend to have more apical dendrites and fewer basal dendrites (Bannister and Larkman, J Comp Neurol 1995).

Anyway, to address this point further we have examined the Soltesz archive in neuromorpho.com containing a set of 3 superficial + 6 deep CA1 pyramidal cells. As far as we know, this is the only annotated dataset available. We aimed to evaluate whether biased morphological properties could account for any effect of phase-locking preference. Unfortunately, the dataset was not robust enough to capture different morphological features (see Scholl diagram below; mean \pm SD). Yet, we ran a subset of simulations, but as expected, we found no morphological effect given similarities between groups.

However, we would like to stress again that our analysis identify morphological features as significant factors. Thus, while we understand that a more exhaustive morphological analysis is required, we feel our data support significant trends. We have revised the text accordingly. See pages 6, 10 and 12.

3. Figure 2G-H. Do the tests of significance account for multiple comparisons being made? It seems they should. If they don't then what justification is there for this?

We always include posthoc statistical confirmation of significant effects validated by multivariate tests. Corrections for multiple comparisons to avoid type I errors are only required if the null hypothesis is to be rejected (that is the case when groups are similar). In those cases, the Bonferroni corrected p-value is always reported to aid in interpretation. Please, note that data in Fig.2F and G was analyzed with the Pearson correlation. We have clarified this whenever necessary in the text.

4. Why are firing phases for the example cells in Figure 3A (all inputs) different to the phases for the same cells in 2F. The effects of morphology that were suggested for 2F-H don't obviously appear to hold up for the 'All inputs' group in Figure 3A.

This is a different individual. We tried to use different sets to provide more examples. Please, note that some individuals are more flexible than others are when expressed in different morphologies, as it is shown in Fig.2F. This is due strong effects of some intrinsic properties that make them firing more or less reliably. We now make the point more clearly in legend and text.

Regarding Fig.3A, the effect of morphology is marginal for this individual. Please, note that the effect of morphology for all inputs (black) in seen independently in Fig.3B (n127 and n409 have both deep- and sup-like behavior, versus n128 y sup1 which are more consistent with sup-like). See statistic report. Also, note

5. The DREADD experiment presented in Figure 4E shows neuronal activity in the presence of CNO, but I couldn't find an analysis to show that CNO has any effect on the neurons relative to control conditions. There are some example histograms in Figure S6 but they are not accompanied by any further analysis. If there is no statistically robust effect of CNO then the results are hard to interpret.

We now include more data in Supp.Fig.6 to clarify this point. Please, note that the DREADD experiment is challenging in that stability may be compromised by i.p injections (animals tend to shake upon injections). Thus paired analysis before and after CNO was only possible in a subset of cases. See experimental design in Supp.Fig.6B (note that the more stability-demanding juxtacellular recording was obtained only after CNO for this reason).

We managed to record stable LFP signals derived from multisite silicon probes before and after in n=5 mice. We found significant decrease of theta power (paired t-test $p=0.016$) and similar laminar profiles of LFP and CSD theta cycles with respect to SLM (no effect of phase reversal; $p=0.527$). Units recorded with multisite probes were sorted again using kilosort2, a toolbox that allows accounting for potential drifts before and after CNO injections. We could reliably isolate n=9 pyramidal cells, n=4 putative PV basket cells and n=2 units classified as bistratified- or OLM-like cells (also expressing PV and the inhibitory DREADD as shown in Fig.3C) before and after CNO. Data support effect of CNO over the firing rate of putative PV+ interneurons (paired t-test $p=0.037$). For pyramidal cells, we found contrasting effects on firing rate across sublayers precluding global statistical effects. Finally, as predicted by our simulations, population level phase-locking behavior evaluated after CNO showed unimodal distribution along the falling theta phase (Fig.3E) in contrast to control data. We now better explain this in the text. See page 8.

6. Overall the manuscript is very dense, uses a lot of jargon and is often hard to follow. As just one example, a "firing reservoir" sounds impressive, but I don't think I have any idea what this is or how it would be "repurposed".

We have revised the text to clarify and avoid jargon.

The figure legends are also hard to follow. As an example, Figure 4H begins "Experimental data confirmed simulation prediction". It would be helpful to refer to the prediction (Figure 4B?) and to explain why the experimental data meets the prediction (it's not clear that it does) and if any tests were carried out to show this is not a chance result.

We thank the reviewer for noting this. The text has been clarified after including new analysis in Fig.4.

Minor

Line 92-93, please state the test used for the multivariate analysis. What are the 'statistical effects for sublayer'? I couldn't find details of the analysis for this data in the methods.

We now add details on the statistics as required. See Fig.1G caption and text page 4

Line 95, "superficial cells tended to fire preferentially..." to what extent is this observation statistically robust?

We support the claim statistically. Fig.1G and I.

Line 97 "concurred" -> "were observed"?

Done

Figure 1F. What is the meaning of the open and closed symbols?

This is now better clarified. Fig.1F and G

Is Figure 1H necessary? I'm not sure what it adds that's useful.

This is now removed to accommodate new data on optotagged superficial and deep cells (see Fig.1H,I).

Reviewer #2 (Remarks to the Author):

The authors combined electrophysiological and computational tools to investigate phase-locking of pyramidal cells in CA1. They found experimentally that superficial cells tended to fire preferentially at the theta trough and deep cells at the theta peak. Then they focused on modeling to better understand what determines this effect. They found effects of dendritic properties, but also strong effects of perisomatic inhibition. To confirm predictions of the simulation they conducted more experiments and manipulated GABAergic population. They found that the GABAergic population can indeed constraint phase selection. They also modeled the influence of CA3 and ECIII inputs using a large regression model and concluded that "internal oscillatory dynamics of CA1 pyramidal cells is determined by intrinsic and microcircuit factors that cooperate to segregate firing across sublayers".

Overall, this study combines experimental and modeling work in a very advanced way. The model is rather sophisticated and builds on top of previous modeling studies using biophysically realistic parameter values. The way the experimental part and the model complement one another is quite remarkable.

We thank the reviewer for positive comments.

My main concerns are in the experimental aspect, which is essential for the rest of the study. The evidence for the main result that superficial cells tended to fire preferentially at the theta trough and deep cells at the theta peak comes from a relatively small number of cells (12 from mice and 28 from rats). It wasn't clear to me if each subpopulation of cells was statistically significant on its own or they were only significant when taken together. From figure 1G that shows phase preference for each cell, the only strong regularity seems to be that mouse superficial layer cells fired at trough. Is this phenomenon driving the entire effect?

Phase preference modulation for each juxtacellularly recorded cell was evaluated with the mean vector length and tested for significance with the Raleigh test ($p < 0.05$ in all cases). We made this point clear in page 3. At the population level, we noted bimodal distributions in both datasets (head-fixed, freely-moving) and tested morphologically reconstructed cells with a multi-sample circular test for preparation and sublayer. We now fully report details from statistical tests in text (page 4) and Fig.1 caption.

Finally, to increase the number of observations and to further extrapolate results we performed a new set of head-fixed experiments using two transgenic lines and micro-LED optoelectrodes allowing for optogenetic tagging of deep and superficial CA1 pyramidal cells independently. Different theta phase preference of deep and superficial opto-tagged pyramidal units was confirmed again in this new independent dataset. See Fig.1H,I and associated text (page 4).

The significance of the findings could be justified more. It is not surprising that different inhibitory and excitatory inputs can change the phase preference of a downstream cell. The work the authors did in identifying some mechanisms that have particularly prominent effect on this is certainly important. However, the significance of these results could be amplified if they could connect these mechanisms with circuit models of hippocampal processing. At the moment, it is not clear what could be a functional significance of their results.

We appreciate this comment. We have reinforced the significance of our predictions, together with the new data analysis included in Fig.4. See pages 10 and 11. We have also highlighted significance along Discussion.

The phase is reported relative to theta oscillations measured at the stratum pyramidale. What is the difference in the phase of theta for deep and superficial cells local theta and does this difference have impact on the model?

This is a relevant question. There was no difference between SP-SLM phase reversal values from deep and superficial cells within each data set (head-fixed and freely-moving). However, we found about 34° difference between preparations. The preparation effect may be due to the use of a contralateral SLM electrode in freely moving versus head-fixed preparation. Importantly, phase-locking dynamics is all reported against the local SP theta, which is similar for cells recorded from deep and superficial sublayers (right). We clarify this point in the revised version. Page 4 and new Fig.S2F

Minor comments:

It is not visible what are the 97 factors that were used in regression shown in figure S7A.

We have redesigned Supp.Fig.7. We also include the full list of all factors in the method section. See page 21

Figure 1G is very important, but it was little hard to read (circles are largely overlapped, making it difficult to evaluate the strength of the effect). Perhaps the authors could visualize this important results with an additional figure to make it more accessible.

We made circles larger and tried to better clarify caption.

How many cells did you analyze in the first section of the results, 12 from mice and 28 from rats or 10 from mice and 20 from rats (line 83 vs line 94)?

We report first data from all juxtacellular cells recorded: 12 cells from 12 mice; 28 cells from 28 rats. In this dataset we noted bimodality of phase preference at the population level (all cells together in each preparation; Fig.1C,D). Next, we evaluated the effects of preparation and cell-type specificity in a subset of these cells that were morphologically reconstructed: 10 cells from 10 mice and 20 cells from 20 rats. We make this point more clearly in the revised version. See page 4.

Reviewer #3 (Remarks to the Author):

In the current manuscript, Navas-Olive et al. use in vivo recordings to evaluate the precise relationship between the phase of hippocampal theta and the firing rate of CA1 pyramidal cells. They observe that most individual neurons have a unimodal relationship of firing rate to theta phase, but that the population displays a bimodal relationship, with some neurons preferentially firing at the trough of theta and others preferentially firing at the peak. The authors then use biologically realistic computational models modified by genetic algorithms to explore the underlying cellular mechanisms which may govern the phase-relationship of firing. They observe a complex interaction between morphology, inhibitory inputs, and excitatory inputs which largely explains the observed differences in phase locking. The authors demonstrate that these inputs can be dynamically altered to modify firing timing and perform some experimental validation of their computational findings.

In general, I find this to be an excellent scientific study and I am very enthusiastic about it. While it is true that bimodal firing with respect to hippocampal theta has been observed in prior studies, here the authors provide highly detailed and compelling arguments which help explain the underlying biological basis for this phenomenon.

We thank the reviewer for positive comments about our work.

I do have a few major and several minor issues with the manuscript that need to be addressed prior to publication.

Major Issues:

1) Many of the figures are poorly described, leaving the reader to guess what the authors are showing. The following list is NOT all-inclusive. The authors should go through each figure and ensure that every item in every panel is fully explained.

We have followed your suggestions and revised captions and text to make figures easier to follow.

Figure 1A: There are four mini panels with only the following description in the figure legend: "Example of a deep CA1 pyramidal cell recorded and labeled juxtacellularly from a head-fixed running mouse." While an electrophysiologist might know that the upper right panel is the average waveform of the recorded neuron's action potentials and the panel below that is an autocorrelogram showing a clear refractory period, many readers will not, and this information needs to be included in the figure legend.

Done

Figure 1B: The LFP traces are not described in the figure legend at all. Are the LFP traces filtered, or raw? Is the reader supposed to be able to see something in those raw traces? It seems like the traces are far too zoomed out to see how the spikes align with the LFP traces.

LFP traces are raw. We have selected a shorter segment to facilitate the reader can track spike timing against LFP. This is now specified in the revised caption.

Figure 1F: What are the solid vs. open circles? What is black vs. green?

This refers to deep (open) and sup (filled) cells. Caption is clarified and legend highlighted

Figure 1G: The figure legend says this is multivariate analysis. This might be what accounts for the p-values in the bottom right of the panel, but it is not clear what is actually being plotted in the primary figure of that panel. I presume that figure is the phase of peak firing for each neuron (with mean and SEM), but the authors should not make the reader guess. Also, when the multivariate analysis finds statistical effects of “preparation”, does that mean there is a difference between rats and mice? If so, I encourage the authors to use the term “species” or “rat vs. mouse” rather than “preparation.” If not, please explain what is meant by “preparation.”

Yes, this refers to a Harrison-Kenji test, equivalent to a circular two-way ANOVA for preparation and sublayer. Caption and text clarified. Regarding the effect of preparation. We considered whether this reflects a species effect but our data advocate more for preparation. Here preparation refers to head-fixed or freely-moving, which are known to involve different circuits (i.e. vestibular function, head-direction signaling, optic flow, etc...) and actually connect with our simulation results showing how different input pathways may bias phase preference. We made a note on this point both in results section (page 4)

Figure 2G-H: What do the dots represent and what data are the box-and-whisker plots based on? I presume the dots are the values for each individual for the four morphologies. Was each individual only simulated once per morphology? If so, and if I am interpreting the graph properly, are the box-and-whisker plots based on only those four datapoints? If so, I'm not sure that the plots are correct. For example, individuals 12 and 15 have what appear to be three datapoints near 360 and one point near 180, but the errorbar extends all the way to ~450. If those are typical box-and-whisker plots, and they are using only the four plotted datapoints, they are not correct. If, on the other hand, the box-and-whisker plots are drawn from other data (perhaps many simulations of each individual/morphology pair), that should be made clear in the figure legend. And if the dots represent the average phase preference for each morphology across many simulations of each individual, this should also be made clear. All of these points apply to Figure 2H as well.

Each dot Fig.2G represent data from one synthetic cell, i.e. individual (set of genes) in one morphology. The number of dots per individual varies from 3 to 4 to meet inclusion criteria (individuals with realistic firing rate values in at least 3 out of 4 morphologies). Hence, box plots for each individual depict the distribution of n=3 or 4 synthetic cells each. In Fig.2H, data are shown for each morphology. The number of synthetic cells in each morphology varies depending on inclusion criteria: n127 (n=10 synthetic cells), n409 (n=11), sup1 (n=11) and n128 (n=12). We have clarified this in the revised caption.

Figure 4E: If I am interpreting the figure correctly, I believe that “...the matrix on the right...” is actually referring to the matrix on the left-hand side of that panel.

Right, we apologize for confusion. The new Fig.4 is modified to accommodate new analysis and to improve narrative.

Figure 4H: What is yellow vs. orange? I presume that is juxtacellular vs. multisite, but this should be clear.

This refers to data from cycle tSC1-2 (high CA3; yellow) versus tSC3-4 (high ECIII; orange) in Fig.4E. It is now clarified in the legend.

Supplemental Figure S8: The description of panel A mentions numbers in parentheses. But I don't see any numbers in parentheses in panel A.

We apologize for this. Numbers are now included.

2) The data presented in Figures 3D and 3E needs to be quantified for statistical significance. It is true that those two graphs appear similar, but the authors should quantify whether the phase locked firing in Figure 3E is statistically more similar the prediction in Figure 3D than 1) chance (using shuffles); and 2) the phase locked firing from control cells (such as the data from Figure 1D).

We have run additional analysis as suggested. First, we tested experimental and simulation phase-locking data and found they are similar (circular ANOVA, $p=0.711$). Second, phase-locking firing under CNO is significantly modulated as tested by Rayleigh and therefore is different from chance by definition. In any case, we tested the preferred phase distribution histogram by Rayleigh and founds it was significantly modulated as well. Finally, phase-locking firing under CNO is statistically different from control data in superficial ($p=0.0089$) and deep cells ($p=0.05$). We have included all these results in the revised version. See page 8 and Fig.3E caption.

3) In addition, much of the data in Figure 4 requires statistical quantification. Are the shifts observed in panel 4E significantly different between deep-like and superficial-like? Is the shift of either group (or both groups) significantly different from chance? The same analysis should be done for the data in panel 4I.

We have run additional analysis as requested. First, data from each cell/unit was contrasted against 1000 surrogates and only those meeting significance (Rayleigh test) for both tSC1-2 and tSC3-4 firing histograms were further considered (Fig.4F). Second, we restricted analysis to theta cycles with consistent coactivation of multiple units recorded with multisite probes, as tested with a rank order test against 500 surrogates Fig.4G). Finally, statistical differences are reported for deep and superficial cells in Fig.4H. See also page 11.

The highlighted sequences in panel 4J also need quantification. It seems to me that for all three of the highlighted examples, the authors could have just as easily drawn diagonal lines in the opposite direction and hit just as many neurons. If the authors want to say that those are “dynamic rank order changes”, they need to show that those particular patterns are unlikely to have arisen by chance. Indeed, showing only three questionable examples with no population summary analysis indicates that this final claim is not well-supported in the data.

This was an example for discussion. After running sequence analysis with the rank order test, we have found novel effects of input pathways and microcircuit structure that may require additional experiments. We therefore chose to remove the panel from this version. We thank the reviewer for highlighting this point, which have taken us to new and interesting observations.

4) It is well-established that experience-dependent firing can induce both short- and long-term changes in synaptic strengths. Given that the authors see that the relative contributions of CA3 vs. ECIII inputs can impact the preferred firing phase, I would like the authors to discuss how they foresee experience-dependent plastic changes in the circuit affecting phase-spiking relationships. For example, can these findings explain the requirement for experience in theta sequence expression (Feng et al., J. Neurosci. 2015)?

We thank for this suggestion. We have now included discussion on this matter in the revised text. See page 12

Minor Issues:

5) There are many grammatical errors throughout the paper that need to be corrected. Here is a short, NOT all-inclusive list:

Abstract: Theta oscillations play *a* major role ...

Introduction, 1st paragraph: At the core of *the* declarative memory system...

Introduction, 2nd paragraph: "CA1 sublayers" should be "CA1 pyramidal sublayers".

Page 4: ... using *morphological* reconstructions...

Page 4: Hodgking-Huxley should be Hodgkin-Huxley

Page 4: To *constrain* the parametric space...

Page 7: Dedicated GABAergic microcircuits *constrain* phase ...

Page 7: ...across individual cells we *ran* a multinomial ...

All corrected. We also asked a native English researcher to read the ms for grammatical errors

6) Figure 2 has no Panel E.

Corrected. Thanks!

7) Identifying sleep based on curled posture and lab notes seems like it would be challenging to replicate. The quantification of sleep needs to use a more replicable method. For example, sleep could be quantified as periods of no movement for >30 seconds.

We actually evaluated videos and looked for as periods of immobility, lasting at least 60 sec, and typically associated with curled-up postures as evaluated from the video. This is now clarified (page 23)

8) For the analysis of possible sleep-based changes in preferred firing phase (Figure 1F, for example), the authors do not separate sleep into SWS vs. REM. However, REM typically accounts for only a small percentage of sleep, and according to Mizuseki et al. (Nat. Neuro., 2011), the shift in preferred phase occurs during REM. Thus, changes in phase preference from wake to sleep could be obscured if all sleep stages are combined. The authors state that head-fixed animals did not display much REM, but if the authors also have sleep recordings from freely moving animals, these sleep periods should be separated into SWS vs. REM (on the basis of delta/theta ratios) and re-analyzed.

Sleep analysis was performed for rats and it refers to REM sleep periods only. This is now clarified in text and caption. We found only some REM shifting cells in our juxtacellular database. We mention this point in the text. See page 3.

9) In the Methods, the authors should denote how many WT vs. PV-Cre mice were used to ensure that the data is not skewed by the near-exclusive use of one or the other genotype.

We now include the number of WT and transgenic mice used for each line independently

10) In the Methods, the authors constrain the firing rate from each model pyramidal cell to 0.1-5Hz. I am unsure how the authors performed this constraint. Did they randomly remove or add spikes until the firing was within the appropriate bounds? Did they remove cells that did not have inherent firing rates within these bounds? This needs to be made more clear.

We consider realistic firing rate values (0.01-8Hz). Same number of cycles for all simulations (1000 cycles per simulation and cell). Clarified in text (page 6) and methods (page 20)

In addition, on page 5, the authors say they chose only individuals with firing rates between 0.01-8Hz (this is repeated in the legend for Supplemental Figure S3). I'm confused as to whether the authors used 0.01-8 Hz or 0.1-5 Hz for all analyzes or whether some analyzes used one firing rate range while other analyzes used a separate range. This should be made clear. Furthermore, all analyzes should use the same firing rate range unless there is a specific scientific reason not to.

We apologize for this mistake. Corrected. Firing range is 0.01 – 8 Hz.

Reviewers' Comments:

Reviewer #1:

Remarks to the Author:

The manuscript is improved in parts but in my view weaknesses remain. Some of these appear substantial but others may simply result from the way the data are presented. The manuscript remains a challenge to read. It's easy enough to get a superficial appreciation of what the authors would like to conclude but it took a great deal of reading and re-reading to assess whether the data support the conclusions. This could perhaps be addressed by rewriting of the manuscript and moving some of the supplemental figures to the main body. I've tried to restrict my comments only to areas I previously commented on and to new parts of the manuscript.

Major points.

1. For the most part, the results presented in Figure 1 are convincing. However, for the optotagging experiment in Figure 1H it's unclear how the analysis distinguished directly from indirectly activated neurons. This seems critical to interpretation of the experiment. The responses in the inset appear to show trial to trial variability. This might not be consistent with the assumed direct activation of the ChR2 expressing neurons. If the responses are indirect then the experiment doesn't really add anything. Related to this issue, how was the spread of viral infection assessed? Was there any expression in CA2 or CA3?
2. The explanation of the genetic evolution strategy could still be clearer. The possible and sampled parameter space should be stated precisely. The text states there were > 16 free parameters. Please state how many exactly and how many possible values each parameter could take. Given these numbers it would be helpful to then give an estimate of the number of possible parameter combinations and the number of combinations that were tested. This should also be made clear for the synaptic parameters. Does "more than 4000 intrinsic parameters" mean parameter combinations or something else?
3. The manuscript now provides some evidence for independent validation of the model "Importantly, the resulting synthetic cells (i.e. combination of intrinsic genes, synaptic genes and morphology) accounted well for other non-fitted intrinsic properties of CA1 pyramidal cells, such as spike afterdepolarization and sags (Fig.S3B; right)". However, at present the evidence takes the form of an example trace. This would be much stronger if quantified and compared directly to experimental data.
4. Figure S3 is important for understanding the manuscript. Could it be included as a main figure?
5. I still couldn't figure out from the manuscript whether there really is an effect of morphology on phase preference. Does analysis of the left panel in Figure 2G with a 1 way ANOVA suggest a significant effect of morphology? Does a 2-way analysis with morphology and ion channel composition as factors reveal any interactions? If I've understood it correctly, Figure S3D implies that morphology is not sufficient to explain phase preferences. At best there is a slight trend to bimodality n127 and n409, but this does not explain the different phase preferences of superficial and deep neurons in Figure 1. At least in my viewer, the labels in the right panels of Figure 2F-G are impossible to read (and therefore to comment on).
6. The CNO experiment remains problematic. A before and during CNO comparison of theta properties and activity of putative interneurons is now provided. However, I could not find a before and during CNO evaluation of theta phase preference of putative pyramidal cells. This seems essential to interpretation of the results. I appreciate that there is now a comparison to the predicted control distributions, but this is not itself particularly strong evidence. A further potential issue is that no control data for possible off-target effects of CNO are reported, e.g. CNO applied to animals expressing mCherry rather than hM4Di-mCherry. The conclusion that "phase-locked dynamics become indistinguishable across sublayers under CNO" is not supported by the analysis. This really does require that one shows they were distinguishable in the same experiment before CNO.

7. The analysis in Figure 4E-H is particularly hard to follow. I think this would be better presented as a separate figure with additional panels to explain how the result in Figure 4H was derived. As I understand it, the analysis calculate the firing phase separately for 'High Ca3' and 'High EC' cycles and then reports the difference. As well as showing the difference it would help to appreciate the results if the values in each high Ca3 and high EC cycles are also shown separately for deep and superficial cells. It would also be helpful to include example to data to help understand the claim that "High EC inputs had effect in significantly advancing spike timing of most superficial cells with respect to high CA3 inputs ($49.8 \pm 63.2\sigma$, different than 0σ , $p=0.019$), as expected for theta phase precession dynamics."

8. Throughout, results of statistical tests should be accompanied by degrees of freedom and the test statistic.

Minor

There are many parts of the text that remain confusing or ambiguous. I'm highlighting only a few examples.

Lines 46-47, What are "high-level flexible dynamical representations"?

Lines 56-57, "Hence, place cells become assembled to earlier and late cells in the behavioural sequence by plasticity mechanisms". I don't understand what this means. Should "cells" be replaced by "locations"?

Line 79 and elsewhere. What is meant by a "reservoir"? I'm imaging something to do with reservoir computing, but I'm not sure there is any evidence for this here? Or something that is stored in excess? Whatever is intended, I strongly advise against using this as it lacks meaning. If it must be used then I recommend to include a clear definition when first used.

Line 94. The double negative is confusing. Could "neither" be removed and "nor" replaced with "or"? Or could "not" be removed?

Line 148. By "different angles" do you mean "different approaches"?

Line 181. "on low and high synaptic potentials" Is this referring to low and high amplitude? Or something else?

Reviewer #2:

Remarks to the Author:

The authors have adequately addressed all of my comments. While the overall amount of data is relatively small to make strong conclusions, this study is still a novel and important contribution.

Reviewer #3:

Remarks to the Author:

The authors have addressed my initial concerns and I believe the manuscript is meaningfully improved. In particular, the addition of optogenetic tagging of deep/superficial neurons in Figure 1 adds considerable weight to the authors' arguments.

I have only one minor issue:

The resolution quality of the figures in the main body pdf that I was able to download were very poor and need to be improved before final publication. The quality of the supplemental figures Word doc were good, which suggests that the image quality was lost during conversion from Word to pdf.

Reviewer #1 (Remarks to the Author):

The manuscript is improved in parts but in my view weaknesses remain. Some of these appear substantial but others may simply result from the way the data are presented. The manuscript remains a challenge to read. It's easy enough to get a superficial appreciation of what the authors would like to conclude but it took a great deal of reading and re-reading to assess whether the data support the conclusions. This could perhaps be addressed by rewriting of the manuscript and moving some of the supplemental figures to the main body. I've tried to restrict my comments only to areas I previously commented on and to new parts of the manuscript.

Major points.

1. For the most part, the results presented in Figure 1 are convincing. However, for the optotagging experiment in Figure 1H it's unclear how the analysis distinguished directly from indirectly activated neurons. This seems critical to interpretation of the experiment. The responses in the inset appear to show trial to trial variability. This might not be consistent with the assumed direct activation of the ChR2 expressing neurons. If the responses are indirect then the experiment doesn't really add anything.

Please, note that connectivity between CA1 pyramidal cells is very limited. According to in vitro data only 9 out of 989 pyr-pyr pairs were found to be connected (Deuchars and Thomson 1996) versus zero connections reported so far in vivo (see English et al., Neuron 2017 for >400,000 neuron pairs). Also, indirect excitatory effects were discarded by Stark et al., Neuron 2014 using similar opto-tagging approaches than us. Please, note that in vivo responses are affected by ongoing activities and hence trial to trial variability is higher than in vitro. Also, some of this variability may be explained by feedforward activation on interneurons which may advance or delay pyramidal cell spikes. To avoid misunderstanding we looked for another example using shorter light pulses (100 ms) and clarified the issue in the text. See page 3.

Related to this issue, how was the spread of viral infection assessed? Was there any expression in CA2 or CA3?

We assessed viral expression and localization of probe tracks in all experiments. Please, note our design guarantee that viral infection is restricted to CA1. First, we used a Calb1-cre line and local AAV-DIO-ChR2 injection to restrict expression in superficial CA1 pyramidal cells (see Fig.1H). Because CA2 and CA3 cells **do not** express Calbindin there is no potential confusion. Granule cells from the dentate gyrus are not targeted by AAV injection so, no additional off-target effects. Second, the Thy1-ChR2-YFP line has restricted ChR2 expression in deep CA1 with no expression in CA2 (see Fig.1A of Dobbins et al., Brain Res 2018). See a representative confocal image at right. We clarified this point in the methods.

2. The explanation of the genetic evolution strategy could still be clearer. The possible and sampled parameter space should be stated precisely. The text states there were > 16 free parameters. Please state how many exactly and how many possible values each parameter could take.

We now provide a new supplementary table with this information (new Table S1). There is a total of 23 free parameters; 17 of which (2 passive parameters, 9 intrinsic conductances and 6 synaptic conductances) were fitted by GA. Values of the remaining 6 free parameters were chosen randomly from the experimental interval. See Table S2 and S3 for range values taken. We have clarified this further in the text (see page 4) and the method section.

Given these numbers it would be helpful to then give an estimate of the number of possible parameter combinations and the number of combinations that were tested. This should also be made clear for the synaptic parameters. Does “more than 4000 intrinsic parameters” mean parameter combinations or something else?

Please note that GA factors can move in a range (see these ranges in Tables S2 and S3). Therefore there are an infinite number of possible combinations. After fitting by GA we end up with a set of values (valid combinations) that fit the behavior for a given morphology (e.g. n128). This is the example we provide in the text. We have rephrased this part and slightly modified Fig.S3A to make the approach more clear. See page 4.

3. The manuscript now provides some evidence for independent validation of the model “Importantly, the resulting synthetic cells (i.e. combination of intrinsic genes, synaptic genes and morphology) accounted well for other non-fitted intrinsic properties of CA1 pyramidal cells, such as spike afterdepolarization and sags (Fig.S3B; right)”. However, at present the evidence takes the form of an example trace. This would be much stronger if quantified and compared directly to experimental data.

We provide an additional table (Table S4) showing model and experimental values of some additional non-fitted intrinsic and oscillatory properties. Model values were within experimental in vitro and/or in vivo ranges. Only mean vector length values were higher than experimental data. We report these values in any case. See page 4.

4. Figure S3 is important for understanding the manuscript. Could it be included as a main figure?

We appreciate this suggestion but after careful consideration we chose to keep the current layout for the sake of the narrative

5. I still couldn't figure out from the manuscript whether there really is an effect of morphology on phase preference. Does analysis of the left panel in Figure 2G with a 1 way ANOVA suggest a significant effect of morphology?

A 1-way ANOVA for morphology does not reach significance: $F(11,32)=1.9$, $p=0.0715$. A non-parametric analysis with the Kruskal-Wallis test showed similar results ($\text{Chi}^2(11)=19.6$, $p=0.0503$). Please, note that the effect of morphology becomes significant in interaction with some ionic channels (see next).

Does a 2-way analysis with morphology and ion channel composition as factors reveal any interactions?

A 2-way ANOVA with morphology and ionic channels as factors revealed interactions for the following set of conductances: type A ($F(23,9)=11.3$, $p=0.0003$), type C ($F(28,2)=26.4$,

$p=0.037$), type L calcium channel ($F(12,24)=26.4$, $p=0.037$), HCN channel ($F(9,28)=2.8$, $p=0.017$), type M ($F(11,25)=2.2$, $p=0.044$) and the leak channel ($F(9,28)=2.8$, $p=0.017$). We report these results in the revised version. See page 5.

If I've understood it correctly, Figure S3D implies that morphology is not sufficient to explain phase preferences. At best there is a slight trend to bimodality n127 and n409, but this does not explain the different phase preferences of superficial and deep neurons in Figure 1.

Fig.3D (now Fig.3E) was meant to show some examples. As we explain in the ms and support with the additional statistical analysis, variations in the distribution of synaptic inputs and intrinsic properties in interaction with morphological features are critical in determining firing dynamics across CA1 sublayers. We have reinforced the message to make it clear. See page 5, last sentence of the section.

At least in my viewer, the labels in the right panels of Figure 2F-G are impossible to read (and therefore to comment on).

We have reorganized the panel to increase the label size. Also, this revised version is uploaded with figures separately from the text to improve figure resolution.

6. The CNO experiment remains problematic.

We are sorry the reviewer still has concerns. We have run additional experiments and analyses to fully address your comments regarding this part. In particular, we recorded another PV-hM4D(Gi) mouse injected with CNO to track the same units before AND after. We also ran additional experiments and analysis of PV-hM4D(Gi) mice injected with vehicle (DMSO). Finally, to exclude CNO off-target effects we obtained new data and performed new analysis of wild-type mice injected with CNO. See next.

A before and during CNO comparison of theta properties and activity of putative interneurons is now provided. However, I could not find a before and during CNO evaluation of theta phase preference of putative pyramidal cells. This seems essential to interpretation of the results. I appreciate that there is now a comparison to the predicted control distributions, but this is not itself particularly strong evidence.

We have run one more experiment to increase the number of pyramidal units recorded **continuously** before and after CNO in PV-hM4D(Gi) mice in order to provide an statistic validation of the phase effect **in exactly the same cells**. Please, note that because CNO affects the firing rate and theta phase has to be estimated from units meeting Rayleigh criteria, only $n=12$ units were included in this paired analysis. As already confirmed in comparisons with control distributions, we found significant effects before and after **CNO in PV-hM4D(Gi) mice** (Fig.S6E; $F(1,22)=4.5$, $p=0.046$). Individual units phase shifted consistently (Fig.3F; circular two paired sample test, $t(11) = 5.7846$, $p = 0.0048$). In addition, we also evaluated theta-phase firing probability distribution from all putative pyramidal units isolated independently before ($n=45$) and after **vehicle** ($n=47$) **in PV-hM4D(Gi) mice** (i.e. same penetration pre and post but different units; Figure S6F). The subset of individual units tracked before and after ($n=45$) did not phase shift ($t(44) = 0.21$, $p = 0.83$). We also found similar sublayer effects before ($F(1,46)=4.3$, $p=0.04$) and after vehicle (sublayer effect, $F(1,49)=5.2$, $p=0.02$), indicating that the effect is not associated to any reaction to the systemic injection.

A further potential issue is that no control data for possible off-target effects of CNO are reported, e.g. CNO applied to animals expressing mCherry rather than hM4Di-mCherry.

To address potential off-target effects of CNO alone we have run an additional set of experiments using $n=3$ wild-type mice. See Fig.S6G for similar theta phase firing distribution of units recorded before ($n=27$) and after CNO ($n=32$; $F(1,57)=0.3$, $p=0.6$). Similar sublayer effects before ($F(1,13)=5.3$, $p=0.038$) and after ($F(1,18)$, $p=0.035$). The subset of individual units tracked before and after ($n=25$) did not phase shift ($t(24) = 0.39$, $p = 0.69$).

The conclusion that “phase-locked dynamics become indistinguishable across sublayers under CNO” is not supported by the analysis. This really does require that one shows they were distinguishable in the same experiment before CNO.

This is addressed this above.

We feel that the ms now prove CNO effect from all angles confirming specificity and hope the reviewer is fully satisfied.

7. The analysis in Figure 4E-H is particularly hard to follow. I think this would be better presented as a separate figure with additional panels to explain how the result in Figure 4H was derived.

Following your suggestion we have separated data in two figures. The new Fig.5 includes new panels to better explain the analysis and to illustrate results. This is now clarified (see page 8).

As I understand it, the analysis calculate the firing phase separately for ‘High Ca3’ and ‘High EC’ cycles and then reports the difference. As well as showing the difference it would help to appreciate the results if the values in each high Ca3 and high EC cycles are also shown separately for deep and superficial cells.

Shown in Fig.5F

It would also be helpful to include example to data to help understand the claim that “High EC inputs had effect in significantly advancing spike timing of most superficial cells with respect to high CA3 inputs ($49.8 \pm 63.2\%$, different than 0% , $p=0.019$), as expected for theta phase precession dynamics.”

We now show examples in Fig.5D. Units are ranked according to high CA3 cycles.

8. Throughout, results of statistical tests should be accompanied by degrees of freedom and the test statistic.

Degrees of freedom were included in figure captions of the previous version. We apologize for not having them in the text also and for some missing values. We have revised the text thoroughly.

Minor

There are many parts of the text that remain confusing or ambiguous. I'm highlighting only a few examples.

Lines 46-47, What are “high-level flexible dynamical representations”?

Deleted to avoid confusion.

Lines 56-57, “Hence, place cells become assembled to earlier and late cells in the behavioural sequence by plasticity mechanisms”. I don’t understand what this means. Should “cells” be replaced by “locations”?

Clarified as suggested

Line 79 and elsewhere. What is meant by a “reservoir”? I’m imaging something to do with reservoir computing, but I’m not sure there is any evidence for this here? Or something that is stored in excess? Whatever is intended, I strongly advise against using this as it lacks meaning. If it must be used then I recommend to include a clear definition when first used.

We clarified the issue. We claim that preferred theta phases constitute firing reservoirs used dynamically to build more flexible representations. The idea is that due to connectivity and intrinsic factors single-cells tend to fire at preferred phases (i.e. the reservoir) from which they shift under the dynamical influence of inputs (high CA3 or high ECIII). Please, note this is coming active in the field (see for instance Leibold preprint in bioRxiv 2019.12.18.880583). We clarify this further in introduction (page 2) and discussion (page 8)

Line 94. The double negative is confusing. Could “neither” be removed and “nor” replaced with “or”? Or could “not” be removed?

Done

Line 148. By “different angles” do you mean “different approaches”?

Clarified. We meant single-cell experimental data obtained with different approaches and in different species.

Line 181. “on low and high synaptic potentials” Is this referring to low and high amplitude? Or something else?

Yes, we meant low and high amplitude synaptic responses. Clarified.

Reviewer #2 (Remarks to the Author):

The authors have adequately addressed all of my comments. While the overall amount of data is relatively small to make strong conclusions, this study is still a novel and important contribution.

We thank the reviewer for useful previous comments.

Reviewer #3 (Remarks to the Author):

The authors have addressed my initial concerns and I believe the manuscript is meaningfully

improved. In particular, the addition of optogenetic tagging of deep/superficial neurons in Figure 1 adds considerable weight to the authors' arguments.

I have only one minor issue:

The resolution quality of the figures in the main body pdf that I was able to download were very poor and need to be improved before final publication. The quality of the supplemental figures Word doc were good, which suggests that the image quality was lost during conversion from Word to pdf.

We thank the reviewer for useful previous comments. We apologize for the low resolution images. We have revised figures to improve quality. The revised version is uploaded with figures separately from the text.

Reviewers' Comments:

Reviewer #1:

Remarks to the Author:

My concerns have been adequately addressed. There are a few issues that are not fully convincing but I don't think they are critical to the main conclusions of the study:

Line 120-121. Connectivity in CA1. While I agree with the authors that from the older literature it appears there is little connectivity between CA1 pyramidal neurons, recent work provides good evidence that there is (see Yang et al. PNAS, 2014). This should be cited in apposition to the papers claiming a lack of connectivity. If I were the authors I would want further evidence that the optogenetic activation is direct, but if they are happy otherwise then this is ok.

The use of 'firing reservoir' remains confusing. In the introduction it is a 'preferred phase' whereas in the discussion it is a 'stable attractor'. Readers might appreciate a little more help with this as in neither case is there an intuitive link to a reservoir.

Lines 807/808 and 829/830. Many readers would appreciate more information about how the specificity of viral expression was determined.

REVIEWERS' COMMENTS:

Reviewer #1 (Remarks to the Author):

My concerns have been adequately addressed. There are a few issues that are not fully convincing but I don't think they are critical to the main conclusions of the study:

Line 120-121. Connectivity in CA1. While I agree with the authors that from the older literature it appears there is little connectivity between CA1 pyramidal neurons, recent work provides good evidence that there is (see Yang et al. PNAS, 2014). This should be cited in apposition to the papers claiming a lack of connectivity. If I were the authors I would want further evidence that the optogenetic activation is direct, but if they are happy otherwise then this is ok.

We thank the reviewer for this reference which is now included in the text.

The use of 'firing reservoir' remains confusing. In the introduction it is a 'preferred phase' whereas in the discussion it is a 'stable attractor'. Readers might appreciate a little more help with this as in neither case is there an intuitive link to a reservoir.

We feel the context of firing reservoir as preferred phases which represent stable attractors is clear.

Lines 807/808 and 829/830. Many readers would appreciate more information about how the specificity of viral expression was determined.

This refers to the Method section. We have further clarified.